

Natural Hazards and Earth System Sciences
EGU
Discussions

# Assessment of the 1783 Scilla landslide-tsunami effects on Calabria and Sicily coasts through numerical modeling

Filippo Zaniboni[1,2], Gianluca Pagnoni[1], Glauco Gallotti[1], Maria Ausilia Paparo[1], Alberto Armigliato[1], Stefano Tinti[1]

[1]Dipartimento di Fisica e Astronomia, Università di Bologna, Bologna, Viale Berti-Pichat 6/2, 40127 Bologna, Italy
[2]Istituto Nazionale di Geofisica e Vulcanologia – Sezione di Bologna, Bologna, via Donato Creti 12, 40128 Bologna, Italy

*Correspondence to*: Filippo Zaniboni (filippo.zaniboni@unibo.it)

**Abstract.** The 1783 Scilla tsunami, induced by a coastal landslide occurring during an intense seismic sequence in Calabria (South Italy), was one of the most lethal ever observed in Italy. It caused more than 1500 fatalities, most of which on the beach close to the town where people gathered to escape earthquake shaking. In this paper, complementing a previous work (Zaniboni et al., 2016) focusing on the very local tsunami effects in the town of Scilla, we study the tsunami impact on the Calabria and Sicily coasts out of Scilla. To this purpose we take into account the same landslide geometry considered in the previous study and perform three tsunami simulations, one embracing a larger region with a 50-m computational grid, and two covering the specific area of Capo Peloro, in Sicily, facing Scilla on the western side of the Messina Straits, with even higher resolution (10 m mesh). Numerical results show a very good agreement with the historical observations in Capo Peloro. Moreover, the resulting global tsunami inundation pattern provides a useful hint for tsunami hazard assessment in the Messina Straits area, which is known to be one of the most exposed to tsunami threat in Italy and in the Mediterranean Sea.

## 1 Introduction

The recent catastrophic tsunamis of Sumatra (2004), Japan (2011) and Sulawesi (2018) have raised the interest in such natural phenomena worldwide, including Europe. In the Mediterranean Sea, they are known to be of smaller magnitude than in the Pacific and in the Indian Oceans, but their effects can be as lethal, owing to the high coastal exposure and vulnerability, constantly growing in the last decades as the result of an increased coastal occupation. This poses the need for more detailed assessments of tsunami hazard and of the consequences of tsunami impact, which also implies the need for more accurate numerical simulation tools.

In this framework, the reconstruction of historical events that affected significantly the coasts in the Mediterranean Sea is very important since they can be used to test numerical simulation codes as well as for better estimates of the hazard and risk. One of the most relevant cases in the Mediterranean, owing to the disastrous impact and the unusual availability of plenty of very detailed observations, is the 6 February 1783 landslide-induced tsunami in Scilla. This tragic event, causing the death of more than 1500 people, was the subject of coeval reports, providing details on the devastating effects along the littoral but also on the coastal landslide that was the cause of the tsunami (see Graziani et al., 2006; Tinti et al., 2007; Gerardi




et al., 2008). In a previous paper (Zaniboni et al., 2016) we investigated the tsunami impact close to the source area, that is in the beaches of the town of Scilla, Marina Grande and Chianalea, where most of the tsunami fatalities were counted, see Figure 1. This was accomplished through numerical simulations under the assumption that the causative landslide was purely subaerial.

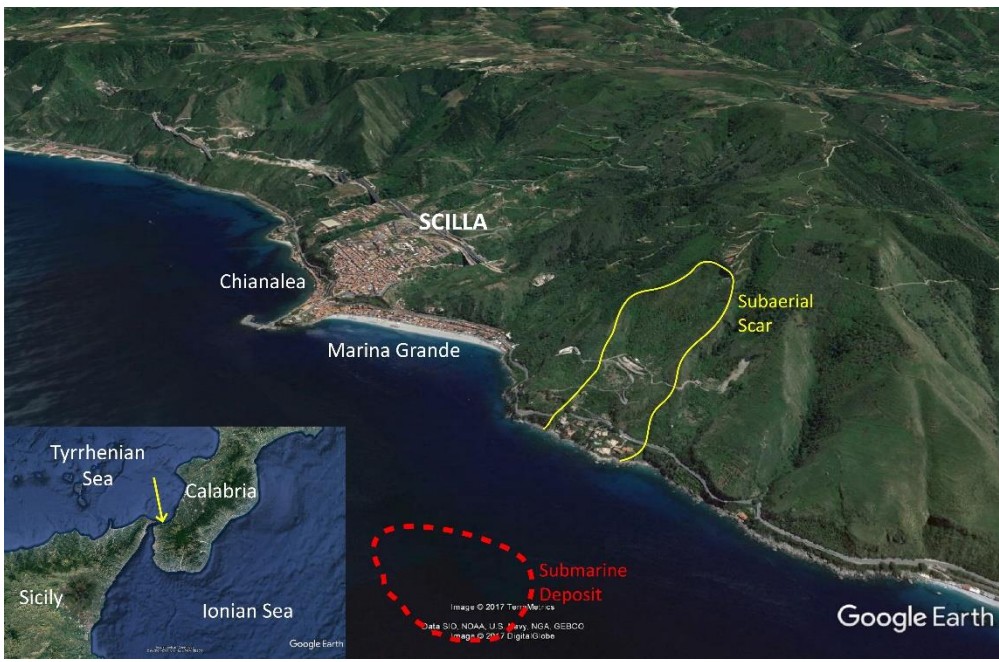

**Figure 1. View of Scilla and its surroundings including the two main beaches of Marina Grande and Chianalea (see the position in the map in the bottom left corner). The yellow contour delimits the landslide scar about 1 km west of the town. The red contour marks the submarine deposit observed in the geo-marine surveys (Bozzano et al., 2011; Casalbore et al., 2014).**

10 As reported by eyewitnesses, tsunami waves affected at least as much as 40 km of the Calabrian coast, the cape of Sicily in front of Scilla, named Capo Peloro, and the north-east coast of Sicily, just south of Messina. In this paper, we carry out two tsunami simulations, one to cover coasts located farther away from the source, and one to study the inundation in the area of Capo Peloro on the opposite side of the Messina Straits. To this purpose, we have performed a preliminary reconstruction of the coeval coastal morphology and topography of Capo Peloro by using features and observations recovered from historical

15 sources. Comparison between numerical results and observations will allow us to evaluate the quality of the simulations.

## 2. The 6 February 1783 landslide-tsunami

In the first months of 1783 a catastrophic seismic sequence, with at least 5 main events, hit the Calabria region (South Italy), provoking about 30,000 casualties and the destruction of over 200 villages (Guidoboni et al., 2007). Cascade effects were



also reported, including tsunamis and diffuse landslide occurrences in the Apennines, that form the backbone of the region (Tinti and Piatanesi, 1996; Graziani et al., 2006; Porfido et al., 2011).

The most tragic event took place during the night of February 6th, in the town of Scilla, on the Tyrrhenian coast of Calabria (see map in Figure 1). The day before, one of the strongest earthquakes of the sequence caused severe destruction in the village, inducing most of the population to find refuge in the wide beach of Marina Grande, west of the town. A strong aftershock in the night caused the collapse of the coastal flank of Mt Pacì in the sea, 1 km west of the Marina Grande beach. The ensuing tsunami killed more than 1500 people (Mercalli, 1906) and destroyed houses and churches (Tinti and Guidoboni, 1988; Graziani et al., 2006).

The huge death toll and the impression raised by the disaster induced the Bourbon government, ruling the region at that time, to finance and arrange several ad-hoc surveys. Further, such a calamity was seen as a big challenge by the coeval international scientific community that was engaged in understanding the nature of earthquakes. Therefore, a number of additional investigations were carried out in the first years after the tragedy by scholars and travelers from different countries (see notes about the mainshock in Guidoboni et al., 2007). The overall result of these observations and research activities that is relevant for our study is that a description of the tsunami source and of the tsunami consequences was made available for future generations (see the discussion in Zaniboni et al., 2016).

In this work, the attention is focused on the tsunami impact outside the surroundings of Scilla. The most relevant reported effects are listed in Table 1 and are mainly gathered from recent papers that quote historical sources (i.e. Graziani et al., 2006; Gerardi et al., 2008; the Italian tsunami catalog by Tinti et al., 2007). The last column includes, among others, the main coeval references. The affected locations are shown in Figure 2.

From Table 1, one can observe that the effects of the tsunami were much less in the neighboring area than in Scilla. This is not surprising since it is typical of landslide-tsunamis to vanish rapidly with distance (see for example Masson et al., 2006; Harbitz et al., 2013, and references therein). However, remarkably the waves were seen in Calabria from Nicotera (point 1, Figure 2), about 40 km north-east of Scilla, to Reggio Calabria (point 5, Figure 2), 20 km south-west. On the other side of the Messina Straits, the place closest to the source is the easternmost corner of Sicily, that is Capo Peloro (point 6, Figure 2), next to the village of Torre Faro (point 7). Here the tsunami was strong and disastrous. In addition to historical sources, the impact of high-energy tsunami waves in this area was confirmed by recent geological investigations carried out in a site called Torre degli Inglesi (Capo Peloro, point 6), about 40 m far from the today shoreline, where a 15 cm thick sand deposit in a trench layer sequence was attributed to the 1783 event (Pantosti et al., 2008).

30



**Table 1. Summary of pieces of evidence of the 1783 tsunami in Calabria and Sicily, with the exclusion of local effects in the area of Scilla.**

| Region | n. | Toponym | Description of the effects | Inundation distance (m) | Run-Up (m) | References |
|---|---|---|---|---|---|---|
| Calabria | 1 | Nicotera | The sea withdrew and then inundated the beach carrying some fishing boats onshore. | | | *De Leone, 1783* |
| | 2 | Bagnara | Affected by the inundation. | | | *Minasi, 1785*<br>*De Lorenzo, 1877* |
| | 3 | Cannitello | Affected by the inundation. | 50 | 2.9 | *Minasi, 1785*<br>*De Lorenzo, 1877* |
| | 4 | Punta del Pezzo | Sea covered the beach by one and a half mile, leaving sand on the ground. | | | *Sarconi, 1784* |
| | 5 | Reggio Calabria | The sea inundated the shore carrying a lot of heavy material. | 80 | 3.2 | *Torcia, 1783*<br>*De Leone, 1783* |
| Sicily | 6 | Capo Peloro | Flooding affected cultivated fields close to the small lake called Pantano Piccolo. Small houses, people and animals were carried seaward.<br><br>Tsunami deposits identified at Torre degli Inglesi. | >400 | 6 | *Augusti, 1783*<br>*Torcia, 1783*<br>*Gallo, 1784*<br><br><br>*Pantosti et al., 2008* |
| | 7 | Torre Faro | Tsunami waves flooded the shore, depositing a large amount of silt and a lot of dead fish. Some boats were carried seaward. 26 people drowned. | | | *Sarconi, 1784*<br>*Torcia, 1783*<br>*Vivenzio, 1788* |
| | 8 | Messina | The sea was seen to rise and to noisily inundate the coast. Waves were also quite relevant at the headlight. Sea level rising by about 2 m, reached the fish market, killing 28 people. | 50 | 2 | *Minasi, 1785*<br>*Vivenzio, 1783*<br>*Spallanzani, 1795* |





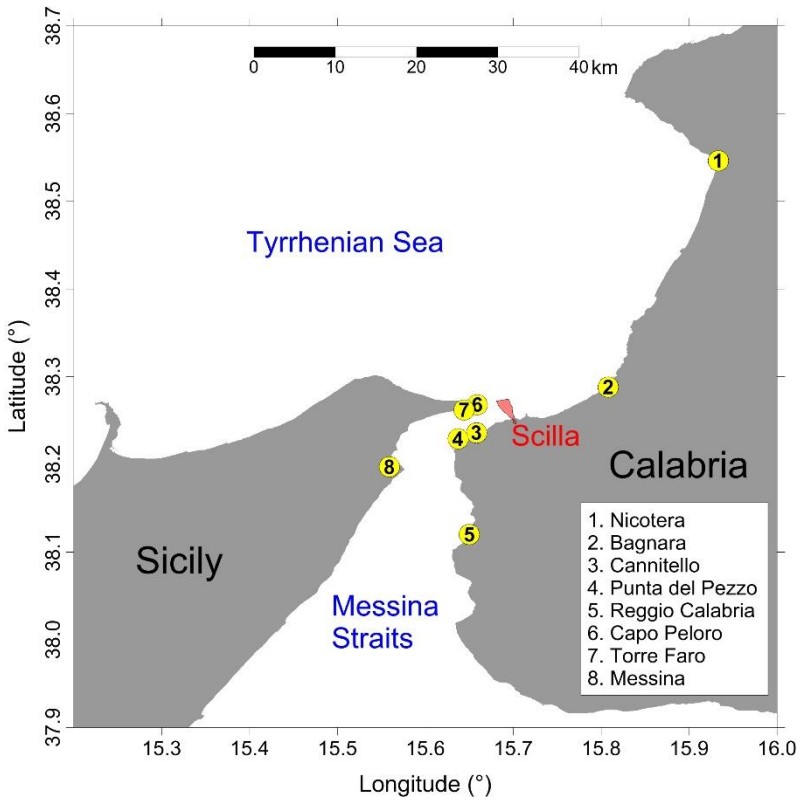

**Figure 2. Toponyms reported in Table 1 (yellow circles). The area swept by the 1783 landslide is marked in red.**

## 3. Numerical Methods

The numerical procedure and techniques used in this paper were developed by the authors and presented in previous applications (see e.g. Tinti et al., 2011; Argnani et al., 2012; Zaniboni et al., 2013; Ceramicola et al., 2014; Zaniboni et al., 2014a, 2014b, 2014c; Zaniboni et al., 2016). Therefore, only a brief description will be given here, since the reader can refer to the quoted papers.

### 3.1 The landslide motion

The numerical code to simulate the landslide motion, UBO-BLOCK1, adopts a Lagrangian approach. The mass is partitioned into a chain of contiguous blocks, and their centers of mass are taken as representative points of the landslide. Blocks are allowed to change shape (length, width, and height, the last one influencing strongly tsunami generation) depending on material rheology, while the total volume is conserved. Bulk (gravity, buoyancy) and surface (bottom friction, drag) forces acting on the blocks are accounted for, together with a block-block interaction governed by some parameters that regulate the sliding behavior during the descent. More details can be found in the aforementioned applications, and in the original description given in Tinti et al. (1997).





## 3.2 Tsunami generation and propagation

The propagation of the waves induced by the landslide moving underwater is modeled through the Navier-Stokes equations in the nonlinear shallow-water approximation and is solved by the code UBO-TSUFD by means of a finite-difference scheme over a regular grid covering the domain of interest. An intermediate code, UBO-TSUIMP, provides a mapping

between the Lagrangian and Eulerian grids (used respectively by the codes UBO-BLOCK1 and UBO-TSUFD). Furthermore, it computes the instantaneous tsunamigenic impulse transferring the sea bottom perturbation due to the landslide to the sea surface by means of an appropriate filter function. This term, time-dependent, is included in the hydrodynamic equations as a source term. The tsunami simulation code accounts also for coastal inundation. This is accomplished by adopting the moving boundary technique (the instantaneous shoreline being identified as the dynamic boundary between wet and dry cells), and

by using a two-flow grid nesting scheme where a higher resolution is needed. This is usually the case in near-shore regions to obtain accurate inundation results. An extended description of the model UBO-TSUFD can be found in Tinti and Tonini (2013).

## 4. The landslide simulation

### 4.1 Scenario reconstruction

The surveys carried out soon after the 1783 event described a wide scar along the Mt. Pacì flank, 1 km west of Scilla, that is still visible today (see Figure 1, yellow contour). Minasi (1785) reported a mass failure starting from 425 m a.s.l. and a landslide front penetrating 100 m into the sea. A later source (De Lorenzo, 1895) mentioned a bigger mass with a larger front (up to 2 km), but a similar offshore mass penetration.

Modern studies performed by means of geophysical surveys and numerical modeling seem to reinforce the hypothesis of a

purely subaerial event, though some elements suggest the possibility of a subaerial-submarine collapse. Interpreting land and submarine survey data, Bosman et al. (2006) proved that the scar continues underwater, at least down to 100 m depth. This feature is reported also in Casalbore et al. (2014) and can be seen as evidence that the landslide involved also a submarine failure. However, it can alternatively be interpreted as the effect of the erosion of the sliding surface by the motion of a purely subaerial mass, which is the picture we adopted in the previous paper (Zaniboni et al., 2016) and we maintain herein.

Further, it is worth mentioning that Bozzano et al. (2011) applied a stress-strain numerical model to evaluate the response of the Mt. Pacì slope to the shaking of the 5th and 6th February earthquakes and concluded that, though the slide scar admittedly extends underwater, the tsunamigenic failure was a purely subaerial collapse, with toe located at about 150 m a.s.l., involving a volume of 5.4 Mm3. They commented that the extension of the scar beyond the initial slide foot might be explained in two ways: either it was the effect of the already cited erosion, produced by the sliding material on the soft

surface sediments nearshore, or it was due to a minor submarine failure that occurred before the main slide. Further, Mazzanti and Bozzano, (2011) using an oversimplified single-block landslide model simulated the subaerial scenario





tsunami. Avolio et al. (2009) used cellular automata to simulate subaerial and submarine landslide dynamics, without paying attention to the generated tsunami.

As regards the landslide deposit offshore, it is sufficient to mention that various submarine surveys found an underwater blocky deposit at about 300 m b.s.l. (see Figure 1, dashed red contour), located less than 2 km far from the coast, and associated it to the 1783 slide. The area covered by the deposit is around 700,000 m2 and can be delimited with a certain degree of accuracy (see Mazzanti and Bozzano, 2011). The total volume is, however, difficult to assess. According to Bozzano et al. (2011), it is compatible with the subaerial scenario landslide; instead, Casalbore et al. (2014) estimated it in the range of 6-8 Mm3.

For the sake of completeness, we mention also that the 1783 Scilla landslide evolution has been the object of a recent application (Wang et al., 2019), where the same subaerial slide scenario has been investigated mainly to test a new numerical method to compute the  landslide motion based on a Eulerian fluid solver, using a second-order central scheme with a minmod-like limiter: in the paper the final deposit was used as a constraint to test the accuracy of the landslide simulations, with no attention given to the generated tsunami.

## 4.2 Results and discussion

In this paper, we make use of the same landslide simulation illustrated by Zaniboni et al. (2016). It is therefore sufficient to synthesize here only the main features of the landslide motion as reported in Figure 3.

The sliding body (upper panel, in blue), reconstructed by considering that the crown is placed at about 400 m a.s.l. and the toe is close to the coastline, results into a subaerial volume of over 6 million m3, with a maximum thickness of about 100 m in the central part. The main parameter governing the motion is the friction coefficient: the back-analysis in view of the comparison between the final simulated position and the observed deposit boundary provided the values 0.23 and 0.06 for the subaerial and underwater portions of the slide motion respectively. The red profile in the upper panel of Figure 3 shows the simulated deposit at about 300 m depth.

The middle and lower panels of Figure 3 show the acceleration and velocity of the CoM of the landslide blocks (black dots) as a function of the distance along the profile, together with the average values (green line). Observe that the initial positive acceleration phase, with values around 3 m/s2, is limited to the subaerial part of the motion (the first 700 m of the profile approximately), while, once underwater, the acceleration adjusts around slightly negative constant values, entailing a slow uniform deceleration phase. This is confirmed by the velocity profile: the average velocity peak is reached just before the entrance of the mass into the water. Notice also that all the CoMs have a similar velocity evolution with distance. The mass has an initial length of around 700 m. For most blocks, the runout is about 2 km, with the exception of the frontal one that travels more than 2.2 km, which results in a lengthening of the slide by about 200 m





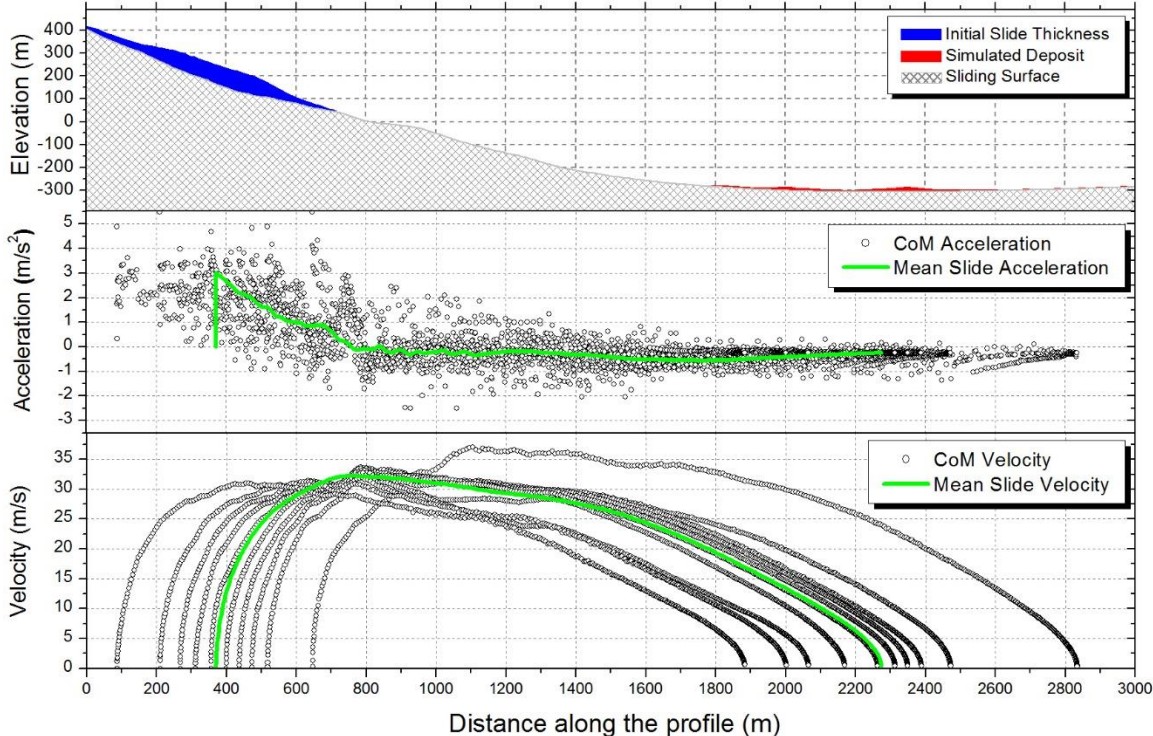

**Figure 3. Scilla 1783 slide simulation by means of the code UBO-BLOCK1. Upper Panel) profile of the initial sliding body (in blue) and of the final simulated deposit at about 300 m depth (in red) over the undisturbed sliding surface (grey area). Middle panel) acceleration of the individual CoMs (black circles) and of the CoM of the whole slide (green line) vs. distance along the sliding profile. Lower panel) CoM velocities (black dots) and average velocity (green line) plotted on the sliding track distance.**

## 5. Tsunami propagation and impact on the coast

The first step of tsunami simulations is the assemblage of the computational grids, the area covered and the adopted grid resolution depending deeply on the goal of the simulation. In general, the higher the grid resolution, the more accurate are the results, but also the heavier is the related computational effort. The tsunami simulations in Zaniboni et al. (2016) were concentrated in the town of Scilla, and the extent of the very local regular mesh was of a few kilometers in length and width with 10-m spaced nodes (see Grid 1, in green, Figure 4).

In this work, our purpose is two-fold, namely: i) to compute tsunami propagation on a wider area, involving tens of km of Calabria and Sicily coasts, and ii) to compute detailed inundation in the specific area of Capo Peloro, located in front of Scilla. For the first goal, we have built a lower resolution grid (50-m space step) denoted in Figure 4 in red as Grid 2, by using the SRTM database (for topography), the GEBCO dataset and nautical charts (for bathymetry). It covers an area of 30x27 km². As concerns the Calabria coast, it runs from Bagnara, 10 km west of Scilla to Reggio Calabria, about 20 km south-westward. Regarding Sicily, the grid includes the city of Messina and 7-8 kilometers of coast southward as well as a piece of the northern coast facing the Tyrrhenian Sea by an extension of at least 10 km.





The choice of Capo Peloro as the target area for the grid resolution enhancement will be justified later, in the discussion of the simulation results obtained on the coarser domain. Figure 4 reports the extent of Grid 3 and 4 (which differences will be cleared out later as well) in blue, that are actually a westward extension of Grid 1, with the same spatial resolution (10 m).

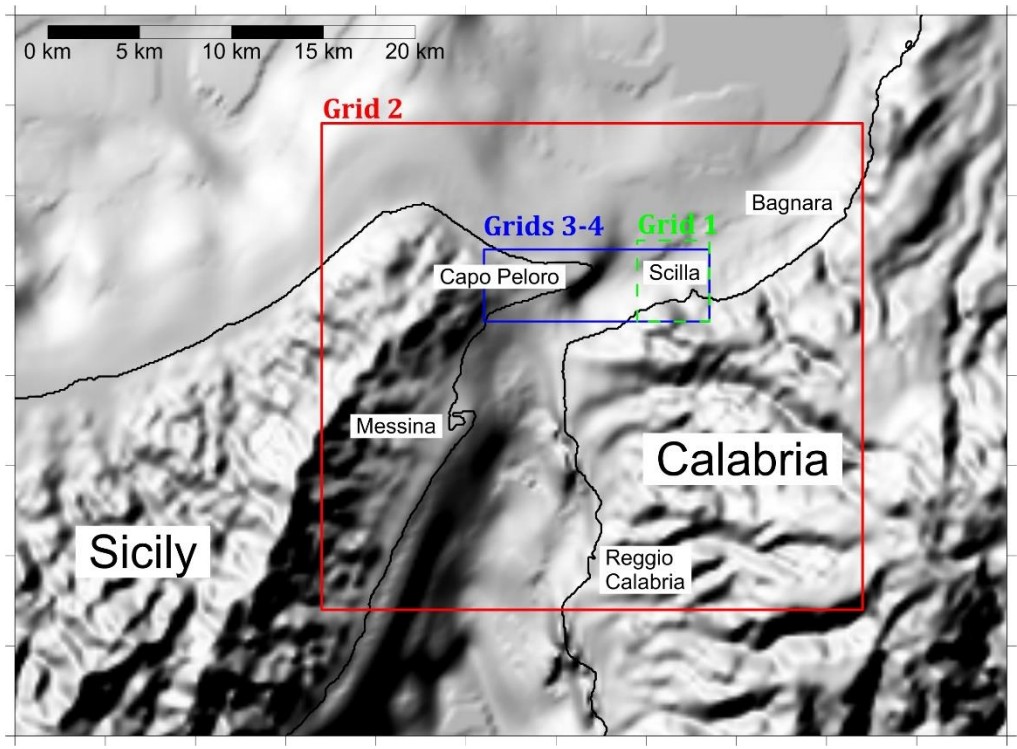

**Figure 4. Computational grids adopted for numerical simulation of tsunami propagation: Grid 1 (dashed-green) was used for the tsunami simulations by Zaniboni et al. (2016); Grid 2 (in red) has been used in the first simulation of this paper; Grid 3 and 4 (in blue) are an extension of Grid 1 covering the target area of Capo Peloro.**

### 5.1 Simulation over the 50 m-spaced grid

Figure 5 shows the maximum water elevation, obtained by considering the maximum value of the sea surface height reached during the simulation time span in each node of Grid 2. This category of plots provides a general spatial pattern of the tsunami energy distribution that, in the intermediate and far-field depends more on the bathymetry than on the source type. Therefore, it allows one to distinguish the areas that are more prone to tsunami attacks.

The area close to the slide shows, as expected, the highest water elevations with more than 8 m in Scilla and waves at least 4 m high (in red) affecting the surroundings coast for about 6 km. The picture of Figure 5 provides also indications on the extension of the coast affected by 1.5-2 m (yellow), and 1 m wave (green), resulting in approximately 10 km and 20 km respectively along the Calabrian coasts. South-westward, in the Messina Strait, maximum height ranges 0.5 m.

Moving to the coast of Sicily, especially north of the Messina Strait, some significant features are observed. First, a strong tsunami energy concentration can be noted towards the easternmost end of Sicily, named Capo Peloro, where waves higher





than 6 m hit the coastline. Along the Tyrrhenian coast of Sicily other tsunami beams are clearly visible (in green, meaning at least 1 m heights), the most interesting of which is the western one, affecting the coastal stretch close to the small village of San Saba. Overall, the simulation confirms the generally known feature that landslide-tsunamis wave height decays rapidly with distance.

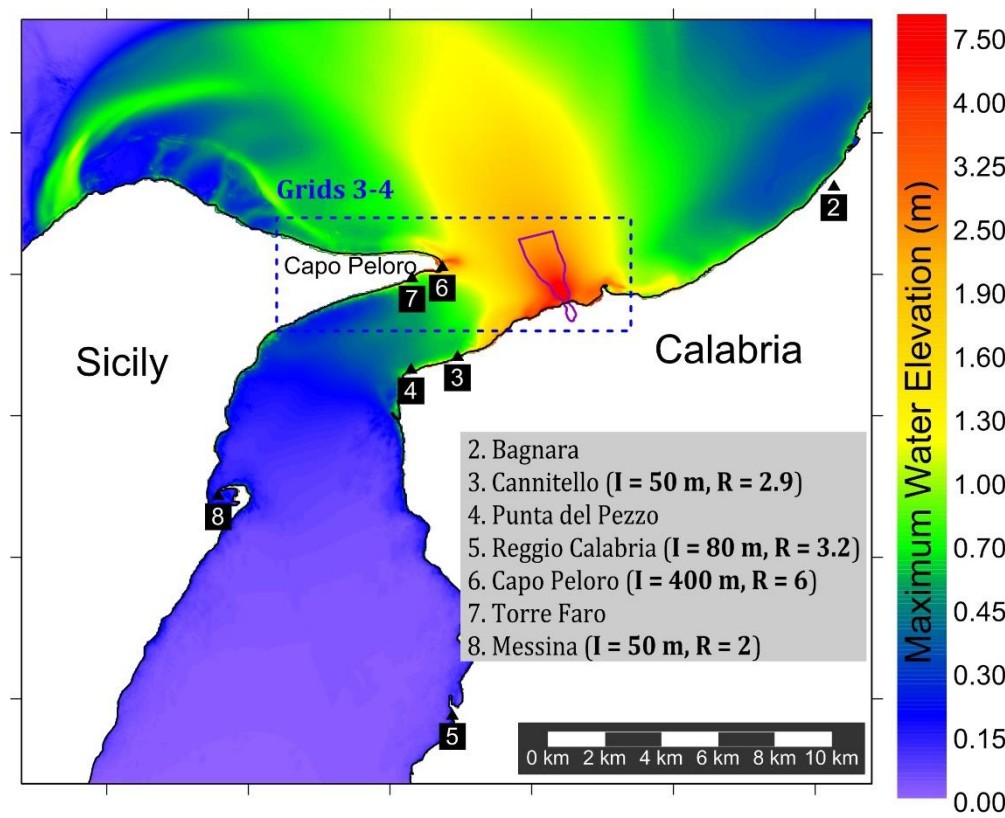

**Figure 5. Maximum tsunami elevation, computed for each point of Grid 2. Numbers represent the positions of the available observations (see Table 1). The respective toponyms are reported in the legend together with the inundation distance (I) and runup (R). The blue dashed rectangle marks the limits of the 10 m resolution Grids 3 and 4, zooming on the area of Capo Peloro, that differ from one another only for a small part of their topography. The purple boundary delimits the sliding surface of the 1783 Scilla landslide.**

When comparing the simulation results with the observed effects a general underestimation can be noticed: for example, in point 3 (Cannitello) of the map of Figure 5 a runup of 2.9 m has been reported, higher than the 1 m maximum height obtained in the simulation. The same holds for Messina (#8) and Reggio Calabria (#5), the most important towns in the Messina Strait, where the difference is by far bigger. Only the area of Capo Peloro (#6 and #7) seems to fit the observations. Such discrepancies can be ascribed to the low resolution of the computational domain (50 meters) not allowing one to describe properly the coastal zone and all the non-linear effects prevailing in shallow water, such as the inland inundation. Further, low-resolution simulations often cannot account for tsunami wave amplification caused by resonances induced




inside coastal basins, such as harbors (see for example Dong at al., 2010; Vela et al., 2010). This latter can explain, for example, the underestimations of the observations reported in Messina and Reggio Calabria.

These considerations push to the use of a more detailed computational grid for the sites of major interest, suggesting that the optimal approach is: 1) to assess the general tsunami energy distribution in the first run of simulations carried out in a wider low-resolution domain 2) to pick up the areas most exposed to tsunami attack (e.g. areas hit by energy beams) to perform higher-resolution investigations. With this approach in mind, the focus has been moved to the area enclosed in the blue-dashed rectangle of Figure 5, including Capo Peloro, where most of the devastating effects of the tsunami outside the source zone were reported (see Table 1, #6 and #7).

## 5.2 First tsunami simulation in the Capo Peloro area

An additional computational grid (Grid 3) has then been built to simulate the tsunami propagation and inundation in this zone.

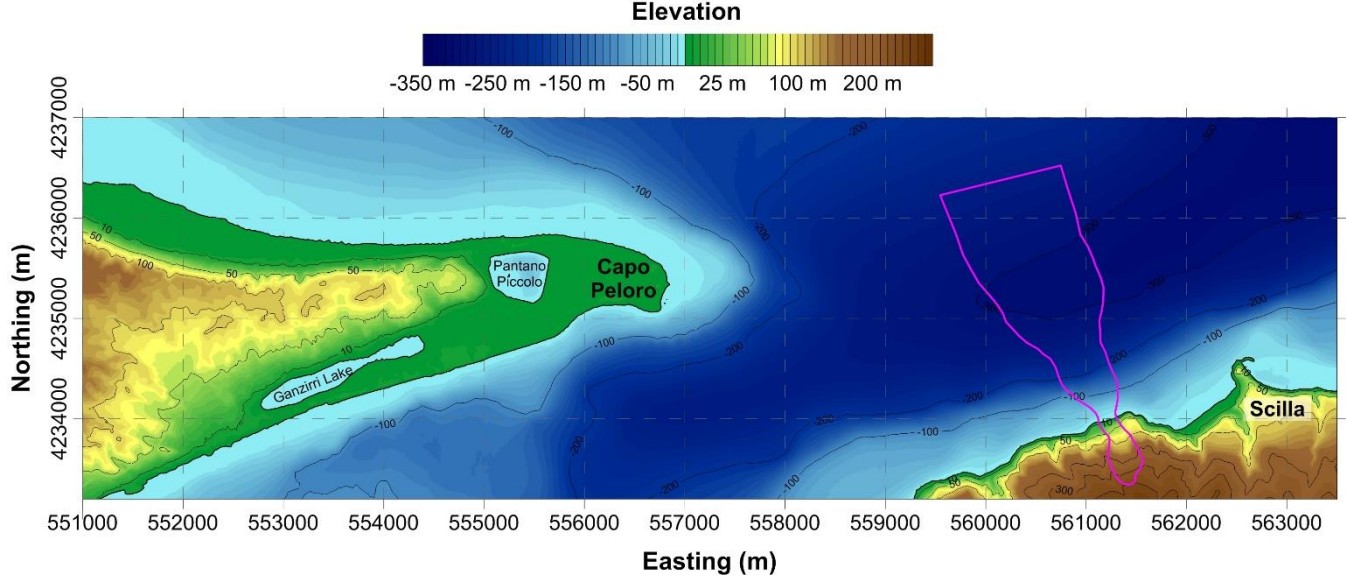

Figure 6. Area covered by Grids 3 and 4 employed to simulate the impact of the 1783 tsunami in Capo Peloro, the easternmost point of Sicily. The magenta contour delimits the strip swept by the landslide.

In this application, actually, due to the proximity of the landslide to the target area, the choice was to build Grid 3 by extending westward the high-resolution grid (Grid 1, 10 m node spacing) used by Zaniboni et al. (2016) for the simulation of the local tsunami effects in Scilla. The relevant data for the Sicilian region was provided by the Civil Protection Department of Sicily and retrieved from the regional cartography service, SITR (Sistema Informativo Territoriale Regionale) in the form of a DTM describing the present topography of the coastal zone with high detail. Grid 3 covers an area of 12.5 km (E-W) by 4 km (N-S), for a total of more than 500,000 nodes. As can be noticed from the map in Figure 6, the Capo Peloro presents a morphology that is radically different from the Calabrian coasts. If in Calabria coasts are steep, rapidly ascending to 400 m



a.s.l., in the area of Capo Peloro a wide lowland is found, extending for about 2x1 km with elevation in the order of 1 to 5 meters. This area is now densely inhabited, with many houses facing the sea especially along the southern coast. Another interesting feature is the presence of two brackish lakes, called Ganzirri Lake and Pantano Piccolo (also known as Faro Lake), characterized by maximum 6 m and 29 m water depth respectively. They are connected to the sea (and to each other)

5   by narrow channels, built under the English authority at the beginning of XIX century (Leonardi et al., 2009; Manganaro et al., 2011; Ferrarin et al., 2013). One of the most relevant observations reported in the historical reconstruction of the tsunami effects is that the basin of Pantano Piccolo was reached by the sea water in 1783 (Minasi, 1785). This feature should be considered in the evaluation of the simulated tsunami accuracy.

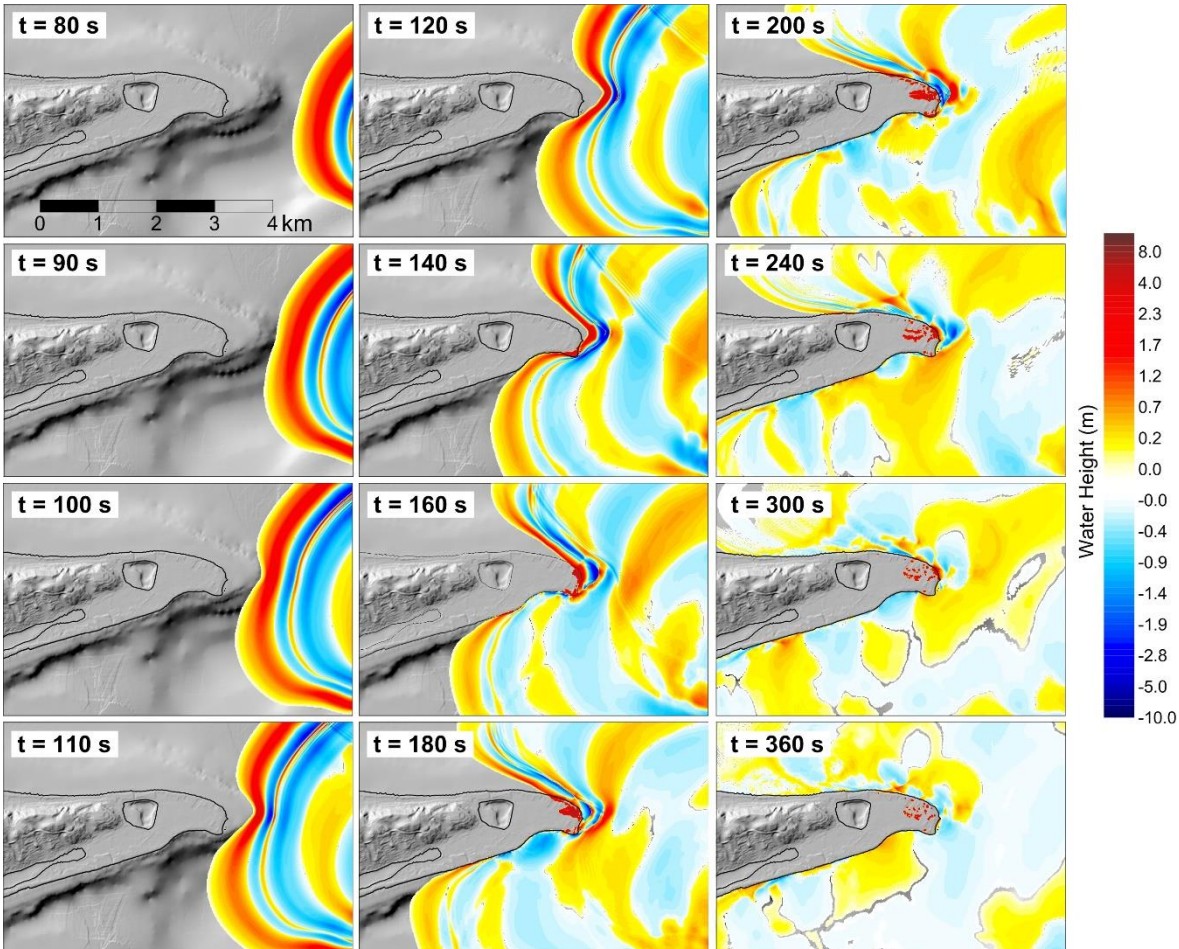

10   **Figure 7. Propagation frames for the 1783 Scilla landslide-tsunami in the target area of Capo Peloro, eastern Sicily.**

Figure 7 shows different frames of the tsunami propagation simulated by means of the code UBO-TSUFD, starting from the snapshot taken at 80 seconds. It is worth pointing out that the origin time coincides with the landslide motion initiation and recalling that, according to our simulation, the slide enters the sea between 10 and 30 seconds, and settles at the final position around 110 s (see Zaniboni et al., 2016, for the accurate description of these results).





The landslide-generated wave heads towards Capo Peloro with a strong positive front 5 m high, directly originated by the slide entering the sea and characterized by an almost circular shape when moving in deep water. At t=100 s the wave begins changing form, due to the interaction with the platform characterizing the seabed east of Capo Peloro. When meeting shallow water, the tsunami is intensely decelerated and begins to deform. Frames at 100 s, 110 s and 120 s display the

subdivision of the wave into two fronts, north and south of the Capo Peloro end, attacking the coast obliquely. At 140 s the southern coast of the cape is reached by the tsunami, that floods the mainland by some tens of meters. The eastern extreme of the area is reached only after, around 20 s later. Here the water penetration is maximum, reaching 600-700 m distance (as visible from 200 s frame). Contemporarily, the northern branch of the tsunami reaches the coastline with a positive 2 m high front that tends to align to a direction parallel to the shoreline. The following oscillations constituting the train of waves

(already visible behind the tsunami front in the first snapshots) do not produce relevant effects on the coast, apart from the cape, where a second relevant positive signal, exceeding 2 m, can be noticed at t=240 s.

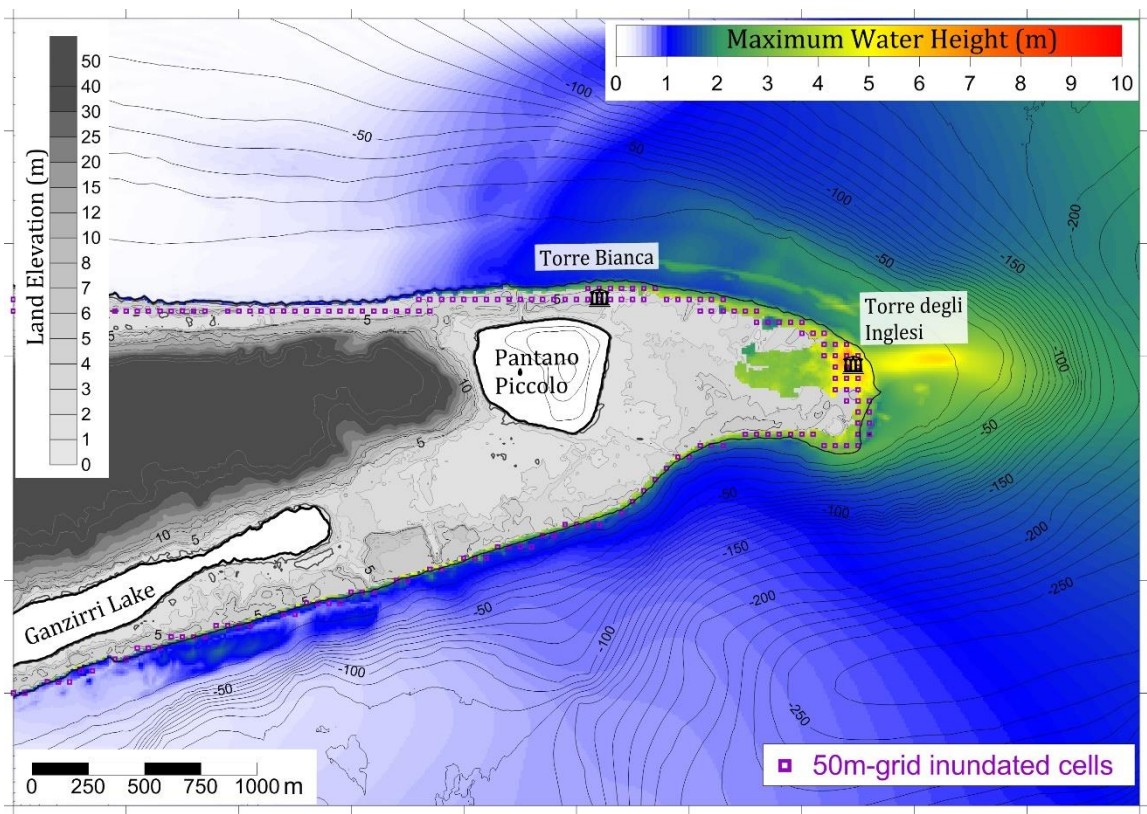

**Figure 8. Maximum water inundation in Capo Peloro for the high-resolution Grid 3 (western portion). The purple squares mark the cells inundated in the simulation with the low-resolution Grid 2. The black symbols represent the positions of historical**

**buildings: Torre degli Inglesi, the site where tsunami deposits were recognized and associated with the 1783 event (Pantosti et al., 2008); Torre Bianca, partially buries by sand deposits (Bottari and Carveni, 2009).**

The maximum water elevation for each node of the western portion of Grid 3 is reported in Figure 8. Some of the features already observed in the 50 m grid (#1) are confirmed, such as the maximum wave height concentration in the easternmost



zone of Capo Peloro. Here, in correspondence with the edifice named Torre degli Inglesi ("English Tower", referring to a building realized during the English domination, marked in black in Figure 8), some tsunami deposits were recognized and associated to the 1783 event (Pantosti et al., 2008). The simulation shows exactly for this spot the maximum water elevation, exceeding 6 m, and the maximum inland penetration, more than 700 m. Furthermore, here the main discrepancy between the

low-resolution Grid 2 and the high-resolution Grid 3 can be noted: in the former case, only a few inland cells (purple in Figure 8) are inundated (corresponding to an inundation distance of 150 m), while in the latter the inland penetration is by far larger. For the remaining coastal stretch, the water flooding is limited to some tens of meters or less, showing little differences between the two simulations. It is very relevant to stress that, concerning the small lake of Pantano Piccolo, the simulations exclude that it is reached by the tsunami, contrary to what reported in the historical reconstructions.

**6. Capo Peloro: morphology reconsideration**

The transition from Grid 2 to Grid 3 produced interesting results and precious insights on the tsunami effects in the target area. Yet, as already noted, one of the most relevant features, i.e. the inundation of the lake of Pantano Piccolo was not reproduced, nor this possibility can fall within the range of uncertainties that are naturally connected with the simulation of natural events like landslides and tsunamis. Indeed, to generate a tsunami able to penetrate further the flat area beyond Torre

degli Inglesi and to cover the distance of more than 2 km separating this from the lake, a much larger landslide should be needed, which is against the geological evidence. Another possibility would be that the tsunami finds a penetration way from other directions, for example from the north. In fact, the strip of land between the basin and the sea is narrow (around 150 m), but is characterized by 3 to 5-meter topography, again really unlikely to be overrun by the simulated tsunami.

A possible alternative solution comes from the considerations on the coastal morphology of this area. In a historical study of

this zone, Randazzo et al. (2014) gave evidence of the strong variability of the coastal stretch around Capo Peloro, due to the action of atmospheric agents. The dataset used to build Grid 3 refers to the present ground level, that can have experienced great changes since the 1783 event. Basing on the historical investigations by Bottari et al. (2006) and Bottari and Carveni (2009), we know that in Pantano Piccolo some hundreds of ships took shelter during the early Roman period (5th century B.C.), and that probably also a small harbor existed, meaning the presence of direct access from the sea. In the following

centuries the channel was filled with sediments, but a look at the topography clearly shows that the lower ground area starts from the north-eastern corner of the lake (as evidenced also by the modern 2-m isoline taken from Bottari et al, 2006), and is the best candidate for the channel that was connecting the lake to the sea more than two thousand years ago. Currently, the access is closed by a 3-4 m sand dune on the coast, that can be reasonably associated to progressive sedimentation due to wind and storms and removed, in order to reconstruct a tentative 1783 morphology of the area.

Another element that needs to be considered is the existence, on the narrow coastal stretch separating Pantano Piccolo from the Tyrrhenian Sea, of a partially buried tower named Torre Bianca ("White Tower", known also as "Mazzone", see Figures 8 and 9 for location), that now is filled with sand up to the first floor (Bottari and Carveni, 2009). Yet in the XVIII century,





this building served as a storehouse, so probably fully available and unburied, meaning that the previously cited assumption of removing 3-4 m of sediments (corresponding to the average height of one floor) in this area is perfectly reasonable.

**Figure 9. Upper panel) Shaded-relief map of the current topography of Capo Peloro, with 2-m isoline (red-dashed line) obtained from Bottari et al. (2006) showing the most likely connection between the lake and the sea, just east of Torre Bianca. The correction done is shown in green-blue when negative (meaning "digging" with respect to the present ground level) and in yellow-red when positive, meaning increased ground elevation. Lower panel) Contour plot of Grid 4, with the low topography from the north-eastern corner of the lake to the sea. In purple, the new 2-m isoline is evidenced. With the corrected topo-bathymetry we have built the Grid 4 that has the same areal coverage as Grid 3.**



Keeping all these considerations in mind, a rearrangement of Grid 3 has been done resulting in a new mesh, the Grid 4, with the same spatial extension. Figure 9 reports in the upper panel a shaded-relief map of the present topography and bathymetry, and the corrections that have been done to reconstruct the likely morphology of 1783. Notice again that in Figure 9 only the western portions of Grid 3 and 4, covering the target area, are reported, since the source area remained

unchanged. Such reconstruction is based on hypotheses and some evidence. For example, the two small channels now connecting Pantano Piccolo to the sea did not exist in 1783, since they were excavated during the following English domination. For this reason, they have been "filled" in Grid 4. The most relevant changes regard the area between the north-east corner of Pantano Piccolo and the Torre Bianca site, where a 1 to 5 m surface layer of ground has been removed. The highest amount of "digging" is located close to the coast, where the sand dune currently closing the possible access of water

(and presumably filling the first floor of Torre Bianca) has been leveled. Morphological changes in other areas of the cape are much less realistic since they would entail lowering of high topography that is less intensely affected by natural agents such as storms, winds or sea waves.

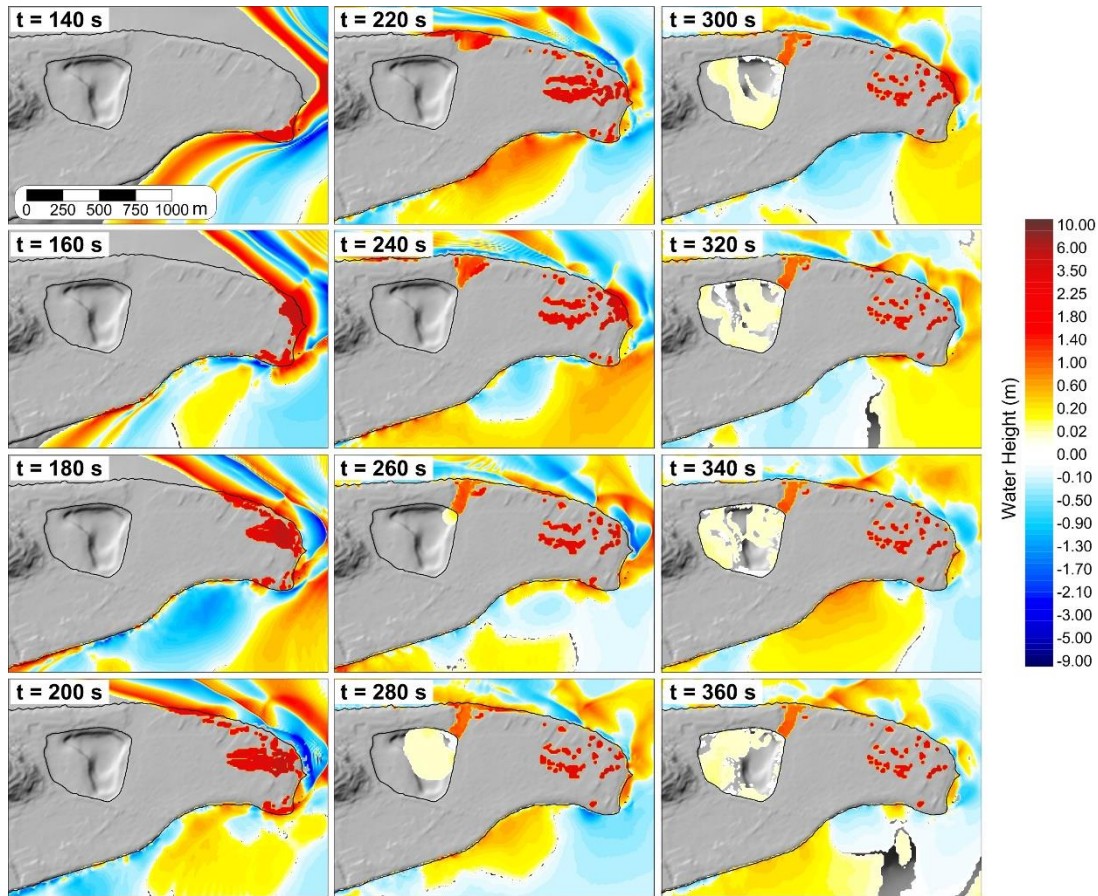

**Figure 10. Propagation frames of the 1783 landslide-generated tsunami on the western part of Grid 4. Notice the water**
**propagation inside the lake.**




The tsunami simulation in Grid 4 coincides with the simulation in Grid 3 for the first 200 seconds, as is shown in Figure 10. The main differences start from the 220 s field when the coastal area east of Torre Bianca begins to be inundated. The following frames show that the water reaches more than 1 m height, channeling through the modified area and reaching the lake between 240 and 260 s. After that, a small positive wave propagating inside the basin can be noticed, 10 to 20 cm high,

5    with also some reflections. The adopted correction to the topography, then, facilitates the tsunami flooding of the Pantano Piccolo lake.

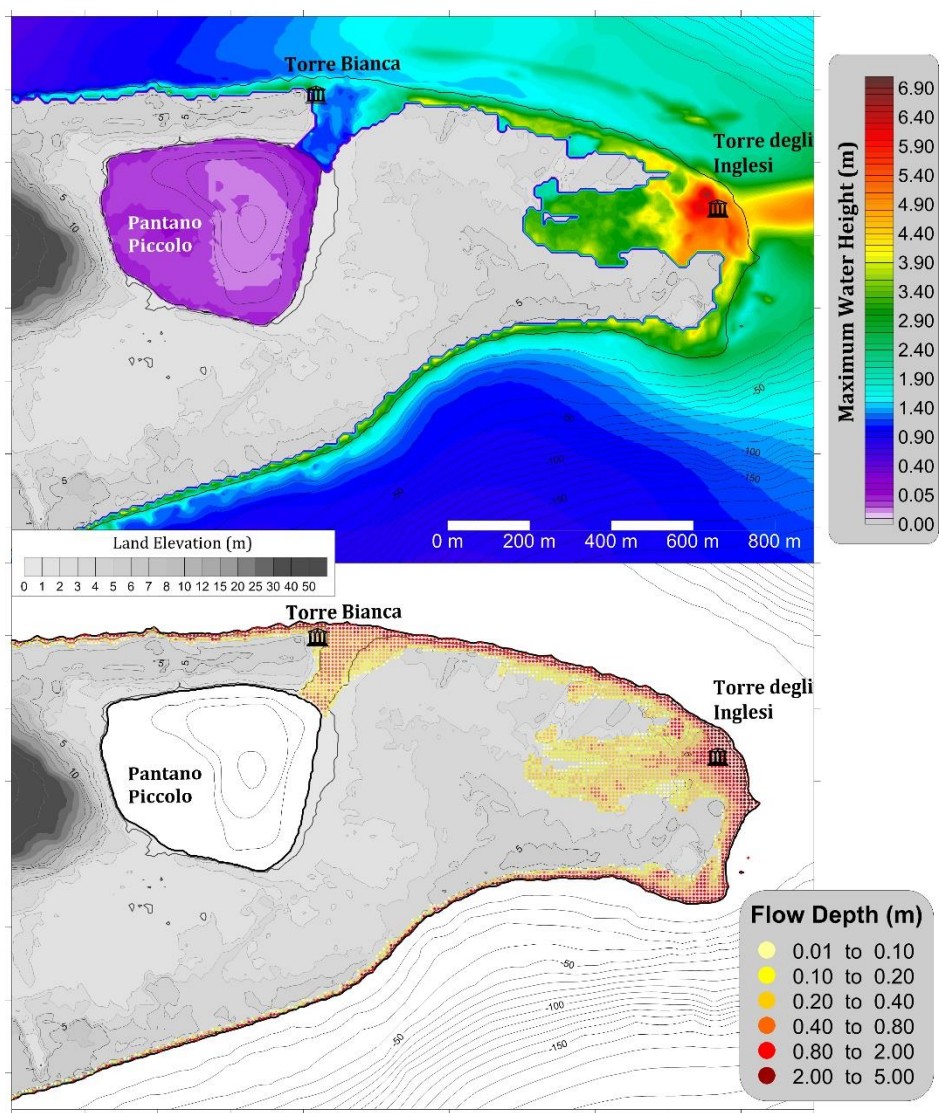

**Figure 11. Upper panel) Maximum water elevation for the western part of Grid 4. Notice that tsunami penetrates also into the lake. Lower panel) Maximum flow depth for each topographic node.**

10    The map with maximum water elevation (Figure 11, upper panel) is analogous to the one of Figure 8. The chief difference comes from the modified zone, where the water climbs up to 2.5 m close to the coast and 1 m just before entering the lake.



Inside the basin, the maximum wave elevation reaches 40 cm in the western part, but less in the central area, that is deeper. The lower panel displays the height of the water column characterizing the flood, known usually as flow depth. The plot shows that the northern coast is affected by higher flow depth, reaching almost everywhere 2 m, with the maximum located at the Torre degli Inglesi (more than 5 m). The flood, just before entering Pantano, is characterized by a water column of about half a meter.

## 7. Conclusions

The work presented here is the natural extension of the Zaniboni et al. (2016) paper, where the initial sliding mass (subaerial) was reconstructed, its descent along the Mt. Pacì flank and its immersion in the sea were simulated, and the effects of the generated wave on Marina Grande beach of Scilla (where most of damages and casualties resulted) were assessed. The tsunami simulations were performed in a high-resolution (10 m) grid, referred to here as Grid 1, covering the town of Scilla (as depicted in Figure 4).

Encouraged by the very satisfactory results obtained, well-fitting the detailed historical reports, in this work we have applied the same numerical codes to complete the study of the tsunami. This has been done by means of additional simulations: one in a wider domain (Grid 2), 30x27 km², covering a larger portion of Calabria and the north-eastern Sicily coasts, characterized by a lower resolution (50 m node spacing); and two in a 12.5x4 km² domain, embracing Capo Peloro, with higher resolution (10 m node spacing) where the present topo-bathymetry (Grid 3) and a reconstructed topo-bathymetry (Grid 4) have been used to investigate tsunami flooding.

The main results and findings are listed in the following.

- The simulation of the landslide motion is by purpose the same as in Zaniboni et al. (2016). The plots of the landslide blocks' accelerations and velocities vs. the distance along the profile show that most of the positive acceleration occurs in the subaerial part of the slip, while the underwater phase exhibits slightly negative, almost constant values, entailing a slow deceleration phase before the sediments deposition at 300 m sea depth.

- The tsunami simulation in the lower resolution Grid 2 (50 m step) provides a picture of the tsunami energy propagation pattern: in addition to the source area, the largest wave heights are reached in the easternmost cape of Sicily, Capo Peloro, where bathymetry causes energy focusing. A further interesting, yet unexpected feature, is the strong tsunami beam reaching the village of San Saba, located along the northern coast of Sicily, 10 km westward. As for the Calabria coast, the wave height rapidly decreases with distance from the tsunami origin, as expected for landslide-induced tsunami events.

- The comparison with historical reports is more satisfactory in the Capo Peloro zone than elsewhere, though the computed inundation does not fit observations. A general underestimation of the wave height is noted, which has to be ascribed to an insufficient description of the local, shallow bathymetry and coastal topography features. The conclusion is that 50-m resolution grid (#2) is suitable to assess the general pattern of the tsunami energy



distribution, highlighting the areas where the highest waves concentrate, while is inadequate to provide good coastal tsunami data for the 1783 case.

- The area presenting the highest interest is Capo Peloro, the easternmost cape of Sicily, where the most damaging effects outside Scilla were recorded, including a deep tsunami penetration and the death of 26 persons, that is the only life toll of the tsunami away from the source. Thus, this zone has been selected as a good target for a grid refinement. A higher resolution computational domain (Grid 3, 10-m spaced) has been then assembled, basing on the present morphology. The tsunami simulation on Grid 3 shows that the inundation distance in the area of maximum impact changes radically, passing from 150 m found for Grid 2 to more than 600 m, in full agreement with the observations. This feature clearly shows the improvement brought by the computational grid adjustment.

- However, the tsunami does not reach the Pantano Piccolo lake, which contrasts sound historical accounts (see Minasi, 1785). This misfit calls for simulation changes that can be done with three possible options: i) to produce a stronger tsunami; ii) to use a better resolution; iii) to change the basic topo-bathymetry dataset, especially in the target area. We have excluded option 1, that is to use Grid 3 and to produce a stronger tsunami by using a larger volume landslide, since the landslide geometry is well constrained by geological considerations on the scar morphology and by the identified offshore deposit, and also by the good agreement with observations of the tsunami run-up height in the Scilla beaches computed through Grid 1 (see Zaniboni et al., 2016). We have further ruled out also option 2, that is to use grid spacing finer than 10 m, since this resolution was enough for the target area, as mentioned before, and also because it was enough to fit the 600-m observed inundation of Capo Peloro with Grid 3. So the only left option was to keep the same resolution and to build another set, Grid 4, that is based on opening a way to the tsunami access through an ancient channel that was a pathway to an inner well-protected harbor in ancient Roman times, and that is presently filled with sediments and obstructed by a coastal sand dune. The signature of this channel is still recognizable today in a linear topographic depression (only 2 m high a.s.l.) going from the north-eastern corner of the lake to the sea. We believe that Grid 4, allowing the tsunami to reach the lake, represents reasonably well the Capo Peloro topography at the time of the 1783 tsunami occurrence, also based on the existence of Torre Bianca building, which existence reinforces the hypothesis of sand deposit removal along the coast. The tsunami reached the lake through the channel either because there was no coastal dune, as assumed in Grid 4, or because the dune was lower than today and was overcome and/or demolished by the incoming tsunami.

- This paper together with the previous paper by Zaniboni et al. (2016) provides a good reconstruction of the 1783 landslide-induced event, respecting the chief geological constraints and fitting the tsunami observations in those places where the tsunami was most lethal, namely the areas of Scilla and Capo Peloro (see simulations in Grids 1 and 4). Still, some level for improvement remains for fitting observations in places covered by Grid 2, which means that probably local high-resolution grids have to be nested to Grid 2 to handle areas where a better agreement has to be searched. But this will be left to further analyses.

A final consideration regards the area of Capo Peloro, that is densely populated with a flat inland topography, and placed in a region with many potential tsunami sources around, either due to earthquakes or to coastal landslides. Hence it needs a particular attention in terms of hazard, vulnerability and risk assessments. This paper, through the simulation carried out on Grid 4, has shown the effect of coastal channels as elements favoring tsunami penetration. Since there are today active channels connecting the lake of Pantano Piccolo and Ganzirri to the waters of the Messina Straits, their role should be explored and taken into account when planning and devising civil protection measures.

## Acknowledgments

The authors are indebted to the Civil Protection Department of the Sicily Region, in the persons of Arch. Giuseppe Marziano and Arch. Biagio Bellassai, for providing the detailed DTM of the eastern Sicily coast; to Dr. Roberto Basili and Dr. Paolo Marco De Martini (INGV, Istituto Nazionale di Geofisica e Vulcanologia, Rome, Italy) for fruitful interactions and discussions about Capo Peloro morphology and tsunami deposit evidence.

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
