# Peer review of "Assessment of the 1783 Scilla landslide-tsunami effects on Calabria and Sicily coasts through numerical modeling"

_Natural Hazards and Earth System Sciences, 2019_

## Referee Comment (RC1) · Marinos CHARALAMPAKIS (Referee) · 17 May 2019

General comments:

The manuscript aims in completing an earlier work based on the simulation of a lethal landslide generated tsunami along the Calabria coast. This paper simulates the tsunami effects in a new region, further away for the landslide source, in Sicily. This is scientifically significant in understanding the tsunami hazard in the area.

The technical approach and the methodology applied are based on commonly approved scientific base and the presentation of the data and the results are clear and

concise. The idea of reconstructing the morphology for better simulating the phenomenon is novel and proved valid.

In order to give the paper a wider approach with a more general appeal, I would suggest discussing more the tsunami hazard and risking assessment issue. More specific, the conclusion of the last paragraph is very interesting and important and it would be nice if it is highlighted more.

The answer is "YES" to all of the following questions, with an exception to question 20. English language can be improved for better understanding:

1. Does the paper address relevant scientific and/or technical questions within the scope of NHESS?

2. Does the paper present new data and/or novel concepts, ideas, tools, methods or results?

3. Are these up to international standards?

4. Are the scientific methods and assumptions valid and outlined clearly?

5. Are the results sufficient to support the interpretations and the conclusions?

6. Does the author reach substantial conclusions?

7. Is the description of the data used, the methods used, the experiments and calculations made, and the results obtained sufficiently complete and accurate to allow their reproduction by fellow scientists (traceability of results)?

8. Does the title clearly and unambiguously reflect the contents of the paper?

9. Does the abstract provide a concise, complete and unambiguous summary of the work done and the results obtained?

10. Are the title and the abstract pertinent, and easy to understand to a wide and diversified audience?

11. Are mathematical formulae, symbols, abbreviations and units correctly defined and used? If the formulae, symbols or abbreviations are numerous, are there tables or appendixes listing them?

12. Is the size, quality and readability of each figure adequate to the type and quantity of data presented?

13. Does the author give proper credit to previous and/or related work, and does he/she indicate clearly his/her own contribution?

14. Are the number and quality of the references appropriate?

15. Are the references accessible by fellow scientists?

16. Is the overall presentation well structured, clear and easy to understand by a wide and general audience?

17. Is the length of the paper adequate, too long or too short?

18. Is there any part of the paper (title, abstract, main text, formulae, symbols, figures and their captions, tables, list of references, appendixes) that needs to be clarified, reduced, added, combined, or eliminated?

19. Is the technical language precise and understandable by fellow scientists?

20. Is the English language of good quality, fluent, simple and easy to read and understand by a wide and diversified audience?

21. Is the amount and quality of supplementary material (if any) appropriate?

Specific comments:

It is not clear how the resolution of the GEBCO grid, which is usually 150m the best, was improved using nautical charts. Which is the resolution of these charts for this quite big area? The concern here is if the re-sampling of the GEBCO grid down to 50 m adds any details or it is just a "cell-split". It might be the case that 50 m bathymetry grid

resolution is needed, just to be at the same level as the onshore topography, which is usually at higher resolution than the bathymetry. If this is so, it should be clearly stated.

With which kind of offshore data the 10 m resolution Grid 3 has been constructed. There is only information for the topography. IF such a resolution is artificial for the offshore region, this should be clearly stated.

The swept area or sliding surface is represented in the figures as a polygon. Who did you define the limits of the area offshore? Was there a detail description in one of the reference papers? Moreover, the bottom limit would look better if it was not a straight line.

Figures 8, 9, 10 & 11 should come after the reference in the text.

A clarification of the terms wave height, wave elevation and flow depth will improve the understanding of the manuscript.

Use constant naming for the grids, e.g. p18 line 32 in contrast to p18 line 24

Parts of the conclusions need rewriting. Some refinement in English language will improve the text. For example, in the last line the word design fits better than "devising", since it is common terminology for this subject.

Technical corrections:

Figure 1: In this figure Capo Peloro and Messina should be indicated in the inset, it should also be mentioned that the yellow arrows points at Scilla. The Google earth image needs indication of the north. Mt Paci should also be pointed in the figure.

P2 line 10: "the cape of Sicily in 10 front of Scilla" it might be more appropriate the term opposite instead of in front.

P3 line 5: "inducing" consider forcing instead

p3 line 7: "ensuing" consider subsequent instead

p3 line 16: "outside" consider along instead

p3 line 21: "vanish" consider attenuate instead

p3 line 24: "corner" consider part instead

p3 line 27: "about 40 m far from the today shoreline" consider "about 40 m onshore, in regard to the present shoreline" instead

Figure 2: Signs for east (E) and north (N) should follow the degree sign in parenthesis for latitude and longitude. The area marked in red is indicated as the landslide swept area. Consider using the tem sliding surface instead.

p6 line 27: "the tsunamigenic failure was a purely subaerial collapse" consider "the tsunami generation was purely attributed to the subaerial collapse" or "the tsunamigenic source was a purely subaerial collapse" instead

p6 line 31: "scenario tsunami" consider "tsunami scenario" instead

p7 line 23: "CoM" initials should be defined, i.e. Center of Mass(?)

p8 line 15: "GEBCO" Which version of GEBCO and at which resolution.

p9 line 2: "reports" considered illustrates instead

P9 line 14: "The picture of Figure 5" consider "The wave height distribution illustrated in Figure 5" instead

p9 line 16: "ranges" you mean reaches?

P10 line 2: "stretch" consider using area instead

Figure 5: "in the legend together with the inundation distance (I) and runup (R)", I would add the word "observed" to avoid any misunderstanding, "in the legend together with the observed inundation distance (I) and runup (R)"

p13 line 7: It seems that the eastern most extreme is reached after 40s (i.e. T180),

although it depends on where you put the limit for eastern extreme.

p13 line 8: "Contemporarily" consider at the same time instead.

p13 line16: consider illustrated instead of "reported"

p13 line 17: "(#1)" consider (#1 figure 5) instead. Is this point #1? It Is not clear.

p14 lines 3-9: Some refinement in English language will improve the text.

p14 lines 11-18: Some refinement in English language will improve the text.

p14 line 21: "agents" consider factors instead

p14 line 22: "Basing" consider Based instead

p15 line 5: "The 5 correction done is shown in green-blue when negative (meaning "digging" with respect to the present ground level) and in yellow-red when positive, meaning increased ground elevation." Some refinement in English language will improve the text.

p16 lin2: "reported" consider shows or illustrates instead.

p16 line7 : "The most relevant changes regard the area between the north-east corner of Pantano Piccolo and the Torre Bianca site, where a 1 to 5 m surface layer of ground has been removed." Consider "The most relevant change regarding the area between the north-east corner of Pantano Piccolo and the Torre Bianca site is the removal of 1 to 5 m of surface ground layer.

p16 line 11: "agents" consider factors instead

p17 line2: "filed" consider frame instead

p17 line 10: "chief" consider main instead

p19 line 4: "outside" consider besides instead

---

## Referee Comment (RC2) · Amos Salamon (Referee) · 30 May 2019

Zaniboni et al. investigate the Scilla tsunami, which is one of the deadliest tsunamis ever occurred in Italy. They simulated the tsunami generated by the subaerial landslide that followed one of the largest aftershocks that occurred during the 1783 earthquakes storm in Calabria, Italy, and compared the results against the historical evidences. Since there were still some disturbing gaps between the computed results and the historical evidences, they improved the grid resolution and restored the topography of the Pantano Piccolo area in northeastern Sicily at the time of the tsunami. This way they were able to match very nicely the computed results with the actual evidences.

[Figure]

This is an elegant investigation, relevant to nowadays tsunami hazard research and evaluation. The study is well constructed, conducted very carefully with much attention to the fine details. The understandings extrapolate the past experience onto the future, emphasizing the hazard posed by tsunamis induced by subaerial landslides, the need to stay away from the coast after strong shaking, and the vulnerability of coastal channels to tsunami penetration inland.

The manuscript is certainly suitable for NHESS, but I would recommend some corrections and improvements before publication as follows:

General Comments

Abstract:

I suggest the authors to state and stress the importance of morphological reconstruction they have implemented in the grid around the area of Pantano Piccolo in order to achieve a better agreement between the high resolution scenario computation and the past evidences.

Conclusions:

Rather than a focused, short and concise, this section is a mixture of results, discussion, summary and some conclusions; it is of the longest sections in the manuscript and hard to follow. For example, the first paragraph is a summary of Zaniboni et al. (2016), the second paragraph is a summary of the present study, the bullets section is "main results and findings" (P. 18 line 18), the middle bullet in page 19 is mainly discussion, etc; and the actual conclusions are spread along this section and hard to follow.

I suggest reorganizing the Conclusions section, it can be divided into discussion and conclusions, or any other useful way; it should be shorter with less repetitions of what have already been said before. It would also be useful to refer back to the relevant figures, and this will help the reader to follow the mentioned issues. At the moment

there is only one as such reference (P. 18, line 11).

I am not an English speaking person, but had the feeling that some language editing is needed.

Technical comments:

P. 1, Line 12: Please consult the editor whether to use a reference in the abstract;

P. 1, line 14: "... three tsunami simulations." while P. 2, lines 11-12 mention two tsunamis...;

P. 1, line 17: Would be more accurate to say 'regional' rather than 'global';

P. 6, line 27: Do you mean 'seismogenic' rather than 'tsunamigenic'?

P. 9, line 2-3: 'cleared out' means to empty, remove, leave, etc... I believe you mean 'will be explained'? If so, please rephrase.

P. 14, Lines 8-9: Please indicate which of the simulations is not in line with the historical accounts.

P. 15 and 17, Captions of Figures 9 and 11: While referring to the upper and lower panel, use colon ":" rather than right side bracket ")".

Figure 1:

The study area is quiet familiar to the Mediterranean and the European communities. However, I would suggest the introducing of an inset that gives a wider geographic orientation for those around the world who are not familiar with this region.

Figures 2 and 5:

Please add the location of Pantano Piccolo and San Saba that are mentioned later on in the text.

Table 1:

There were several tsunamis in Calabria and Sicily during the 1783 earthquakes crisis. Please mention in the caption the exact date of the Scilla tsunami;

Please verify whether the historical sources were careful enough to differentiate the effects induced by the Scilla tsunami from the effects of the other tsunamis (e.g. the February 5, 7, March 1, 28);

In order to get a comprehensive perspective of the impact of the Scilla tsunami, I would suggest to complete the given list. For example, please mention what had happened in Pantano Piccolo, San Saba, and elsewhere if known. In my opinion, it worth mentioning also what had happened in Scilla even though this was already investigated in the previous, 2016 paper;

Punta del Pezzo: Please verify whether the sea affected one and a half mile stretch of the beach. One may think that the sea inundated one and a half mile into the land?

---

## Referee Comment (RC3) · Anonymous Referee #3 · 3 Jun 2019

This manuscript describes new analysis of previous studied 1783 Scilla landslide. The analysis in this paper is focused on the inundation of the generated tsunami in the areas close to the landslide. It also includes a reconstruction of the topography to fulfil the historical observations.

The paper is very interesting, and I liked the open and transparent way the results are presented. Good and clear figures. I have suggested minor revision with some suggestions to improvements below.

Main comments:

1) Discussion of results of simulations on Grid2. Line 16++ on page10: Looking at the

maximum surface elevations of Fig. 5, the main argument for not achieving observed runup at location 5 and 8 is the resolution of the grid. Why not perform grid refinement tests? I guess that the resolution is high enough for propagation in the deepest part of the strait. 0.3 m before runup cannot give 2 and 3.2 m runup.

2) What is the original resolution of the data SRTM and GEBCO used in this paper

3) Grid refinement tests – should be shown or at least referred to for all grids (not only the 50 m grid, but in sea and on land for 10 m grid)

Minor comments

1) Table 1 – must have a ref. to Fig 2

2) Line 1 page 7: what is "cellular automata"

3) L5 p7: check super scripts: m3->mˆ3, m2->mˆ2 etc. Check entire paper

4) L8 p7 vs L18 p7. Inconsistent use of "million" and "M". Check entire paper

5) Fig 3: a vertical line at shoreline will help reading the figure. Include also the location of "blocky deposit", not only simulated deposits

6) Fig 3: Use of only end-paranteses for dividing the text for different panels. I think it is better to use colon (Check rest of paper)

7) L8 p8: Higher grid resolution give more accurate results must be more discussed. Resolution of grids and data, stability etc.

8) L14 p9: the sentence starting with "The picture of . . ." must be revised – I could not understand what was meant here

9) Chap 5.1 – I think some mariograms for 3-4 locations also could be fine for better understand the wave pattern

10) Fig 6: include also depth toward location 5 and 8.

11) L19 p 11: what is meant by "this zone"?

12) L21 p11. Revise sentence "If in Calabria...". Show the lowland of Capo Peloro in a map?

13) L18 p13: What is meant by (#1), similar L32 p18 (#2). Is it "Grid 2"?

14) Fig. 11 upper panel. For better comparison to Fig 8, use same scales!

15) L12-14 p18: Check sentence

16) L30 p18: "inundation does not fit observations"??? See L12 p18 and bulletpoint at L28 p19 and elsewhere where you have concluded that the simulations is a good reconstructions.

17) L6 p19: "basing" use based instead?

18) L12 p19. What is meant by "better resolution" – finer grid or higher resolution of the data

---

## Author Comment (AC1) · 17 Jun 2019

AUTHORS ANSWERS TO REFEREE #1's COMMENTS We greatly appreciate the contribution of this reviewer. Here follow the answers to his very useful comments and remarks. In many of them, the reference to the page and line number of the new version appears. The manuscript in its new version is attached, with corrections and changes marked in red.

General comment: The manuscript aims in completing an earlier work based on the simulation of a lethal landslide generated tsunami along the Calabria coast. This paper simulates the tsunami effects in a new region, further away for the landslide source,

in Sicily. This is scientifically significant in understanding the tsunami hazard in the area. The technical approach and the methodology applied are based on commonly approved scientific base and the presentation of the data and the results are clear and concise. The idea of reconstructing the morphology for better simulating the phenomenon is novel and proved valid. In order to give the paper a wider approach with a more general appeal, I would suggest discussing more the tsunami hazard and risking assessment issue. More specific, the conclusion of the last paragraph is very interesting and important and it would be nice if it is highlighted more. *** ANSWER *** Concerning the tsunami hazard and risk issues, the first has been discussed widely, while the second requires additional studies in order to quantify the impact on population and buildings. We believe it is a subject of great interest, but we prefer leaving it for future work.

Specific comments: It is not clear how the resolution of the GEBCO grid, which is usually 150m the best, was improved using nautical charts. Which is the resolution of these charts for this quite big area? The concern here is if the re-sampling of the GEBCO grid down to 50 m adds any details or it is just a "cell-split". It might be the case that 50 m bathymetry grid resolution is needed, just to be at the same level as the onshore topography, which is usually at higher resolution than the bathymetry. If this is so, it should be clearly stated. *** ANSWER *** The part concerning the available datasets has been improved (page 8, lines 15-19 of the new version of the manuscript). Indeed, we did not use GEBCO, but EMODNET and we complemented it nearshore by means of the nautical chart covering the region of interest to allow for better accounting of local non-linear, effects.

With which kind of offshore data the 10 m resolution Grid 3 has been constructed. There is only information for the topography. IF such a resolution is artificial for the offshore region, this should be clearly stated. *** ANSWER *** Offshore data for Grids 3 and 4 have been retrieved from the same nautical chart digitized for the construction of Grid 2. We added sentence in the text (page 11, lines 27-28 of the new version).

The swept area or sliding surface is represented in the figures as a polygon. Who did you define the limits of the area offshore? Was there a detail description in one of the reference papers? Moreover, the bottom limit would look better if it was not a straight line. \*\*\* ANSWER \*\*\* The slide boundary is one of the inputs provided to the landslide simulation code, and its definition was given more in details in Zaniboni et al. (2016), and not repeated here. It is designed basing: on the observed deposit; on the initial sliding body contour; on assumptions about the mass spreading during the motion.

Figures 8, 9, 10 & 11 should come after the reference in the text. \*\*\* ANSWER \*\*\* The figures have been moved after the text.

A clarification of the terms wave height, wave elevation and flow depth will improve the understanding of the manuscript. \*\*\* ANSWER \*\*\* Added terminology at Page 9, Lines 8-12 of the new version.

Use constant naming for the grids, e.g. p18 line 32 in contrast to p18 line 24 \*\*\* ANSWER \*\*\* Fixed.

Parts of the conclusions need rewriting. Some refinement in English language will improve the text. \*\*\* ANSWER \*\*\* Conclusions have been changed and reorganized.

For example, in the last line the word design fits better than "devising", since it is common terminology for this subject. \*\*\* ANSWER \*\*\* Fixed.

Technical corrections:

Figure 1: In this figure Capo Peloro and Messina should be indicated in the inset, it should also be mentioned that the yellow arrows points at Scilla. The Google earth image needs indication of the north. Mt Paci should also be pointed in the figure. \*\*\* ANSWER \*\*\* Figure 1 has been modified according to these indications as well.

P2 line 10: "the cape of Sicily in front of Scilla" it might be more appropriate the term opposite instead of in front. \*\*\* ANSWER \*\*\* Substituted "in front of" with "facing".

[Figure]

P3 line 5: "inducing" consider forcing instead \*\*\* ANSWER \*\*\* Ok

p3 line 7: "ensuing" consider subsequent instead \*\*\* ANSWER \*\*\* We prefer "ensuing" since the tsunami is a consequence of the slide (Page 3, Line 12 of the new version).

p3 line 16: "outside" consider along instead \*\*\* ANSWER \*\*\* The tsunami effects are studied out of the surroundings of Scilla, in a wider domain.

p3 line 21: "vanish" consider attenuate instead \*\*\* ANSWER \*\*\* Ok

p3 line 24: "corner" consider part instead \*\*\* ANSWER \*\*\* Ok

p3 line 27: "about 40 m far from the today shoreline" consider "about 40 m onshore, in regard to the present shoreline" instead \*\*\* ANSWER \*\*\* Corrected "today" with "modern".

Figure 2: Signs for east (E) and north (N) should follow the degree sign in parenthesis for latitude and longitude. The area marked in red is indicated as the landslide swept area. Consider using the tem sliding surface instead. \*\*\* ANSWER \*\*\* Ok

p6 line 27: "the tsunamigenic failure was a purely subaerial collapse" consider "the tsunami generation was purely attributed to the subaerial collapse" or "the tsunamigenic source was a purely subaerial collapse" instead \*\*\* ANSWER \*\*\* Ok

p6 line 31: "scenario tsunami" consider "tsunami scenario" instead \*\*\* ANSWER \*\*\* Ok

p7 line 23: "CoM" initials should be defined, i.e. Center of Mass(?) \*\*\* ANSWER \*\*\* Correct, added at Page 5, Line 10 of the new version.

p8 line 15: "GEBCO" Which version of GEBCO and at which resolution. \*\*\* ANSWER \*\*\* See answer to the first comment in the previous section.

p9 line 2: "reports" considered illustrates instead \*\*\* ANSWER \*\*\* Ok

P9 line 14: "The picture of Figure 5" consider "The wave height distribution illustrated

in Figure 5" instead *** ANSWER *** Ok

p9 line 16: "ranges" you mean reaches? *** ANSWER *** Ok, fixed

P10 line 2: "stretch" consider using area instead *** ANSWER *** Ok

Figure 5: "in the legend together with the inundation distance (I) and runup (R)", I would add the word "observed" to avoid any misunderstanding, "in the legend together with the observed inundation distance (I) and runup (R)" *** ANSWER *** Correct remark, the word "observed" has been added in the figure caption.

p13 line 7: It seems that the eastern most extreme is reached after 40s (i.e. T180), although it depends on where you put the limit for eastern extreme. *** ANSWER *** Ok, changed (Page 12, Lines 19-20).

p13 line 8: "Contemporarily" consider at the same time instead. *** ANSWER *** Done (Page 12, Line 21).

p13 line16: consider illustrated instead of "reported" *** ANSWER *** Changed (Page 13, Line 7).

p13 line 17: "(#1)" consider (#1 figure 5) instead. Is this point #1? It Is not clear. *** ANSWER *** Indeed, it's Grid 2. Fixed.

p14 lines 3-9: Some refinement in English language will improve the text. *** ANSWER *** Done.

p14 lines 11-18: Some refinement in English language will improve the text. *** ANSWER *** Done.

p14 line 21: "agents" consider factors instead *** ANSWER *** "agents" refers to atmospheric manifestations (rain, wind, storms and so on), that are usually denoted with this word.

p14 line 22: "Basing" consider Based instead *** ANSWER *** Done

p15 line 5: "The correction done is shown in green-blue when negative (meaning "digging" with respect to the present ground level) and in yellow-red when positive, meaning increased ground elevation." Some refinement in English language will improve the text. *** ANSWER *** Done

p16 lin2: "reported" consider shows or illustrates instead. *** ANSWER *** Done

p16 line7 : "The most relevant changes regard the area between the north-east corner of Pantano Piccolo and the Torre Bianca site, where a 1 to 5 m surface layer of ground has been removed." Consider "The most relevant change regarding the area between the north-east corner of Pantano Piccolo and the Torre Bianca site is the removal of 1 to 5 m of surface ground layer. *** ANSWER *** Done

p16 line 11: "agents" consider factors instead *** ANSWER *** Same as above

p17 line2: "filed" consider frame instead *** ANSWER *** Ok

p17 line 10: "chief" consider main instead *** ANSWER *** Ok

p19 line 4: "outside" consider besides instead *** ANSWER *** Ok

Please also note the supplement to this comment:
https://www.nat-hazards-earth-syst-sci-discuss.net/nhess-2019-94/nhess-2019-94-AC1-supplement.pdf
* * *
[Figure]

**Supplement:**

**Assessment of the 1783 Scilla landslide-tsunami effects on Calabria and Sicily coasts through numerical modeling**

Filippo Zaniboni[1,2], Gianluca Pagnoni[1], Glauco Gallotti[1], Maria Ausilia Paparo[1], Alberto Armigliato[1], Stefano Tinti[1]

[1]Dipartimento di Fisica e Astronomia, Università di Bologna, Bologna, Viale Berti-Pichat 6/2, 40127 Bologna, Italy
[2]Istituto Nazionale di Geofisica e Vulcanologia – Sezione di Bologna, Bologna, via Donato Creti 12, 40128 Bologna, Italy

*Correspondence to*: Filippo Zaniboni (filippo.zaniboni@unibo.it)

**Abstract.** The 1783 Scilla tsunami, induced by a coastal landslide occurring during an intense seismic sequence in Calabria (South Italy), was one of the most lethal ever observed in Italy. It caused more than 1500 fatalities, most of which on the beach close to the town where people gathered to escape earthquake shaking. In this paper, complementing a previous work focusing on the very local tsunami effects in the town of Scilla, we study the tsunami impact on the Calabria and Sicily coasts out of Scilla. To this purpose we take into account the same landslide geometry considered in the previous study and perform three tsunami simulations, one embracing a larger region with a 50-m computational grid, and two covering the specific area of Capo Peloro, in Sicily, facing Scilla on the western side of the Messina Straits, with even higher resolution (10 m mesh). In this area, reconstructing the local morphology at the time of the tsunami occurrence allows to account for the inundation of the small lake of Pantano Piccolo, event reported in historical records but not reproduced in the previous numerical simulations. In general, the 
[revised manuscript text omitted]

[Figure]

**Figure 1: The logo of Copernicus Publications.**

---

## Author Comment (AC2) · 17 Jun 2019

We followed most of the suggestions of reviewer#2. The answers to his useful remarks follow, the new version of the paper is attached with changes evidenced in red.

General Comments: Zaniboni et al. investigate the Scilla tsunami, which is one of the deadliest tsunamis ever occurred in Italy. They simulated the tsunami generated by the subaerial landslide that followed one of the largest aftershocks that occurred during the 1783 earthquakes storm in Calabria, Italy, and compared the results against the historical evidences. Since there were still some disturbing gaps between the computed results and the historical evidences, they improved the grid resolution and restored

the topography of the Pantano Piccolo area in northeastern Sicily at the time of the tsunami. This way they were able to match very nicely the computed results with the actual evidences. This is an elegant investigation, relevant to nowadays tsunami hazard research and evaluation. The study is well constructed, conducted very carefully with much attention to the fine details. The understandings extrapolate the past experience onto the future, emphasizing the hazard posed by tsunamis induced by subaerial landslides, the need to stay away from the coast after strong shaking, and the vulnerability of coastal channels to tsunami penetration inland.

The manuscript is certainly suitable for NHESS, but I would recommend some corrections and improvements before publication as follows:

Abstract: I suggest the authors to state and stress the importance of morphological reconstruction they have implemented in the grid around the area of Pantano Piccolo in order to achieve a better agreement between the high resolution scenario computation and the past evidences. \*\*\* ANSWER \*\*\* We agree with this suggestion, a sentence has been added to the abstract.

Conclusions: Rather than a focused, short and concise, this section is a mixture of results, discussion, summary and some conclusions; it is of the longest sections in the manuscript and hard to follow. For example, the first paragraph is a summary of Zaniboni et al. (2016), the second paragraph is a summary of the present study, the bullets section is "main results and findings" (P. 18 line 18), the middle bullet in page 19 is mainly discussion, etc; and the actual conclusions are spread along this section and hard to follow. I suggest reorganizing the Conclusions section, it can be divided into discussion and conclusions, or any other useful way; it should be shorter with less repetitions of what have already been said before. It would also be useful to refer back to the relevant figures, and this will help the reader to follow the mentioned issues. At the moment there is only one as such reference (P. 18, line 11). I am not an English speaking person, but had the feeling that some language editing is needed. \*\*\* ANSWER \*\*\* The Conclusions section has been reorganized following this advice. A

Discussion section has been added, resuming and discussing the main results, while the Conclusions section has been shortened.

Technical comments:

P. 1, Line 12: Please consult the editor whether to use a reference in the abstract *** ANSWER *** The reference has been removed.

P. 1, line 14: ": : : three tsunami simulations." while P. 2, lines 11-12 mention two tsunamis: : :; *** ANSWER *** Ok, fixed.

P. 1, line 17: Would be more accurate to say 'regional' rather than 'global'; *** ANSWER *** Ok

P. 6, line 27: Do you mean 'seismogenic' rather than 'tsunamigenic'? *** ANSWER *** Changed "tsunamigenic failure" to "tsunamigenic source" (Page 6, Line 27 of the new version).

P. 9, line 2-3: 'cleared out' means to empty, remove, leave, etc: : : I believe you mean 'will be explained'? If so, please rephrase. *** ANSWER *** Done (Page 9, Line 6 of the new version).

P. 14, Lines 8-9: Please indicate which of the simulations is not in line with the historical accounts. *** ANSWER *** The sentence is referred to both simulations, on Grid 2 and Grid 3, that are named just before. The expression "the simulations" has been substituted with "both simulations" (Page 14, Line 1 of the new version).

P. 15 and 17, Captions of Figures 9 and 11: While referring to the upper and lower panel, use colon ":" rather than right side bracket ")". *** ANSWER *** Done.

Figure 1: The study area is quiet familiar to the Mediterranean and the European communities. However, I would suggest the introducing of an inset that gives a wider geographic orientation for those around the world who are not familiar with this region. *** ANSWER *** A larger map of Italy, with the indication of the studied area, has been

added.

Figures 2 and 5: Please add the location of Pantano Piccolo and San Saba that are mentioned later on in the text. *** ANSWER *** Figure 2 illustrates the location of the observed effects, while the ones in San Saba are obtained from the numerical simulations. The toponym has been added to Figure 5. The position of Pantano Piccolo is reported in Figure 6. Figures 2 and 5 cover a wider region and adding Pantano Piccolo would create confusion.

Table 1: There were several tsunamis in Calabria and Sicily during the 1783 earthquakes crisis. Please mention in the caption the exact date of the Scilla tsunami; *** ANSWER *** Done

Please verify whether the historical sources were careful enough to differentiate the effects induced by the Scilla tsunami from the effects of the other tsunamis (e.g. the February 5, 7, March 1, 28); *** ANSWER *** The main tsunami evidences (excluding the February 6th, 1783 one) are associated to the stronger earthquake of the day before, producing significant waves on the coasts of Sicily and Calabria. Their effects are very well reconstructed and reported in the paper by Graziani et al. (2006), clearly distinguishing the main features of the different events, basing on detailed historical reports.

In order to get a comprehensive perspective of the impact of the Scilla tsunami, I would suggest to complete the given list. For example, please mention what had happened in Pantano Piccolo, San Saba, and elsewhere if known. In my opinion, it worth mentioning also what had happened in Scilla even though this was already investigated in the previous, 2016 paper; *** ANSWER *** The effects in the Scilla beaches have been recalled in the text (Section 2, describing the 6th February 1783 tsunami effects). Thus we preferred to avoid repetition in Table 1.

Punta del Pezzo: Please verify whether the sea affected one and a half mile stretch of the beach. One may think that the sea inundated one and a half mile into the land?

\*\*\* ANSWER \*\*\* The Calabrian coast is very steep, with narrow beaches and high topographical gradients. So, in that case a similar inland penetration is not possible (see also topography in Figure 6)

Please also note the supplement to this comment:
https://www.nat-hazards-earth-syst-sci-discuss.net/nhess-2019-94/nhess-2019-94-AC2-supplement.pdf

---

## Author Comment (AC3) · 17 Jun 2019

AUTHORS ANSWERS TO REFEREE #3's COMMENTS The suggestions and comments by this referee have been useful, and many of them have been integrated into the manuscript. The new version of the paper, with changes evidenced in red, has been attached.

Main comments:

This manuscript describes new analysis of previous studied 1783 Scilla landslide. The analysis in this paper is focused on the inundation of the generated tsunami in the

areas close to the landslide. It also includes a reconstruction of the topography to fulfil the historical observations. The paper is very interesting, and I liked the open and transparent way the results are presented. Good and clear figures. I have suggested minor revision with some suggestions to improvements below.

1) Discussion of results of simulations on Grid2. Line 16 on page10: Looking at the maximum surface elevations of Fig. 5, the main argument for not achieving observed runup at location 5 and 8 is the resolution of the grid. Why not perform grid refinement tests? I guess that the resolution is high enough for propagation in the deepest part of the strait. 0.3 m before runup cannot give 2 and 3.2 m runup. *** ANSWER *** Grid refinements in the harbors of Reggio Calabria (#5) and Messina (#8) could account for higher waves, associated to the tsunami itself and to resonance phenomena. This very interesting task has not been explored yet, and it is indicated as one of the topics deserving attention for future work.

2) What is the original resolution of the data SRTM and GEBCO used in this paper *** ANSWER *** A more detailed description of the original datasets and of their use to obtain the computational grids have been added (page 8 lines 15-19).

3) Grid refinement tests – should be shown or at least referred to for all grids (not only the 50 m grid, but in sea and on land for 10 m grid) *** ANSWER *** The grid refinement has been done and discussed only for the area of Capo Peloro, showing the improvements brought by such procedure. It can then be applied to other areas of interest, where detailed historical reports are available, such as Messina and Reggio Calabria harbors as well as the coastal area of Punta del Pezzo, about 7 km west of Scilla along the Calabrian coast. As already stated above, this is left to further studies.

Minor comments 1) Table 1 – must have a ref. to Fig 2 *** ANSWER *** Done

2) Line 1 page 7: what is "cellular automata" *** ANSWER *** It's a simulation technique, described and applied in the related paper, that is not worthy of description here.

Interactive
comment

3) L5 p7: check super scripts: m3->mĚĘ3, m2->mĚĘ2 etc. Check entire paper *** ANSWER *** Done

4) L8 p7 vs L18 p7. Inconsistent use of "million" and "M". Check entire paper *** ANSWER *** Ok

5) Fig 3: a vertical line at shoreline will help reading the figure. Include also the location of "blocky deposit", not only simulated deposits *** ANSWER *** A horizontal light-blue line has been added, marking the sea level. A profile of the blocky deposit is not available

6) Fig 3: Use of only end-parentheses for dividing the text for different panels. I think it is better to use colon (Check rest of paper) *** ANSWER *** Ok, done

7) L8 p8: Higher grid resolution give more accurate results must be more discussed. Resolution of grids and data, stability etc. *** ANSWER *** As already mentioned above, these issues have been discussed for the Capo Peloro area. Such details are interesting but too specific for the aims of this work.

8) L14 p9: the sentence starting with "The picture of : : :" must be revised – I could not understand what was meant here *** ANSWER *** The sentence has been slightly modified (Page 10, Lines 7 of the new version)

9) Chap 5.1 – I think some mariograms for 3-4 locations also could be fine for better understand the wave pattern *** ANSWER *** We decided to omit the virtual marigrams obtained by the simulations for two main reasons: i) there are not instrumental records to compare with; ii) they did not add particularly interesting information to the already shown plots (propagation and maximum water elevation).

10) Fig 6: include also depth toward location 5 and 8. *** ANSWER *** This remark is not particularly clear: the picture shows the area covered by Grids 3 and 4, not including the harbors of Messina and Reggio Calabria, located southward.

11) L19 p 11: what is meant by "this zone"? *** ANSWER *** No "this zone" exists at

the indicated line and page. There is "coastal zone" (at Page 11, Line 27 of the new version), that is clearly referred to the Sicilian region, already cited twice in the same sentence and not repeated.

12) L21 p11. Revise sentence "If in Calabria: : :". Show the lowland of Capo Peloro in a map? *** ANSWER *** The Capo Peloro lowland and Calabria coasts are shown clearly in Figure 6.

13) L18 p13: What is meant by (#1), similar L32 p18 (#2). Is it "Grid 2"? *** ANSWER *** To avoid continuous repetition of the word "grid", sometimes they have been referred to only using their number ("#1" is for Grid 1, "#2" for Grid 2 and so on).

14) Fig. 11 upper panel. For better comparison to Fig 8, use same scales! *** ANSWER *** Correct remark, the water height scale and the color palette of Figure 8 in the new version match those of Figure 11.

15) L12-14 p18: Check sentence *** ANSWER *** Done.

16) L30 p18: "inundation does not fit observations"??? See L12 p18 and bulletpoint at L28 p19 and elsewhere where you have concluded that the simulations is a good reconstructions. *** ANSWER *** The three reported sentences refer to three different contexts: the first concerns the 50-m grid simulation, not fitting the observed run-up and then justifying grid refinement; the second refers to the simulation of the 10-m grid in Scilla beaches, presented in the previous paper (Zaniboni et al., 2016); the third one discusses the good agreement reached with the morphology reconstruction adopted for Grid 4.

17) L6 p19: "basing" use based instead? *** ANSWER *** Ok, fixed.

18) L12 p19. What is meant by "better resolution" – finer grid or higher resolution of the data *** ANSWER *** It means a finer grid, and it is explained later when excluding option 2 (Page 19, Line 28).

Please also note the supplement to this comment:
https://www.nat-hazards-earth-syst-sci-discuss.net/nhess-2019-94/nhess-2019-94-
AC3-supplement.pdf

———————————————————
2019-94, 2019.

---

## Author Response (AR1)

**AUTHORS ANSWERS TO REFEREE #1's COMMENTS**

General comment:

The manuscript aims in completing an earlier work based on the simulation of a lethal landslide generated tsunami along the Calabria coast. This paper simulates the tsunami effects in a new region, further away for the landslide source, in Sicily. This is scientifically significant in understanding the tsunami hazard in the area. The technical approach and the methodology applied are based on commonly approved scientific base and the presentation of the data and the results are clear and concise. The idea of reconstructing the morphology for better simulating the phenomenon is novel and proved valid.

In order to give the paper a wider approach with a more general appeal, I would suggest discussing more the tsunami hazard and risking assessment issue. More specific, the conclusion of the last paragraph is very interesting and important and it would be nice if it is highlighted more.

*Concerning the tsunami hazard and risk issues, the first has been discussed widely, while the second requires additional studies in order to quantify the impact on population and buildings. We believe it is a subject of great interest, but we prefer leaving it for future work.*

Specific comments:

It is not clear how the resolution of the GEBCO grid, which is usually 150m the best, was improved using nautical charts. Which is the resolution of these charts for this quite big area? The concern here is if the re-sampling of the GEBCO grid down to 50 m adds any details or it is just a "cell-split". It might be the case that 50 m bathymetry grid resolution is needed, just to be at the same level as the onshore topography, which is usually at higher resolution than the bathymetry. If this is so, it should be clearly stated.

*The part concerning the available datasets has been improved (page 8, lines 15-19 of the new version of the manuscript). Indeed, we did not used GEBCO, but EMODNET and we complemented it nearshore by means of the nautical chart covering the region of interest to allow for a better accounting of local non-linear, effects.*

With which kind of offshore data the 10 m resolution Grid 3 has been constructed. There is only information for the topography. IF such a resolution is artificial for the offshore region, this should be clearly stated.

*Offshore data for Grids 3 and 4 have been retrieved from the same nautical chart digitized for the construction of Grid 2. Added sentence in the text (page 11, line 24 of the new version).*

The swept area or sliding surface is represented in the figures as a polygon. Who did you define the limits of the area offshore? Was there a detail description in one of the reference papers? Moreover, the bottom limit would look better if it was not a straight line.

*The slide boundary is one of the inputs provided to the landslide simulation code, and its definition was given more in details in Zaniboni et al. (2016), and not repeated here. It is designed basing: on the observed deposit; on the initial sliding body contour; on assumptions about the mass spreading during the motion.*

Figures 8, 9, 10 & 11 should come after the reference in the text.

*The figures have been moved after the text.*

A clarification of the terms wave height, wave elevation and flow depth will improve the understanding of the manuscript.

*Added terminology at Page 9, Lines 7-10 of the new version.*

Use constant naming for the grids, e.g. p18 line 32 in contrast to p18 line 24

*Fixed.*

Parts of the conclusions need rewriting. Some refinement in English language will improve the text.

*Conclusions have been changed and reorganized.*

For example, in the last line the word design fits better than "devising", since it is common terminology for this subject.

*Fixed.*

Technical corrections:

Figure 1: In this figure Capo Peloro and Messina should be indicated in the inset, it should also be mentioned that the yellow arrows points at Scilla. The Google earth image needs indication of the north. Mt Paci should also be pointed in the figure.

*Figure 1 has been modified according also to these indications.*

P2 line 10: "the cape of Sicily in front of Scilla" it might be more appropriate the term opposite instead of in front.

*Substituted "in front of" with "facing".*

P3 line 5: "inducing" consider forcing instead

*Ok*

p3 line 7: "ensuing" consider subsequent instead

*We prefer "ensuing" since the tsunami is a consequence of the slide (Page 3, Line 12).*

p3 line 16: "outside" consider along instead

*The tsunami effects are studied out of the surroundings of Scilla, in a wider domain.*

p3 line 21: "vanish" consider attenuate instead

*Ok*

p3 line 24: "corner" consider part instead

*Ok*

p3 line 27: "about 40 m far from the today shoreline" consider "about 40 m onshore, in regard to the present shoreline" instead

*Corrected "today" with "modern".*

Figure 2: Signs for east (E) and north (N) should follow the degree sign in parenthesis for latitude and longitude. The area marked in red is indicated as the landslide swept area. Consider using the tem sliding surface instead.

*Ok*

p6 line 27: "the tsunamigenic failure was a purely subaerial collapse" consider "the tsunami generation was purely attributed to the subaerial collapse" or "the tsunamigenic source was a purely subaerial collapse" instead

*Ok*

p6 line 31: "scenario tsunami" consider "tsunami scenario" instead

*Ok*

p7 line 23: "CoM" initials should be defined, i.e. Center of Mass(?)

*Correct, added at Page 5, Line 10 of the new version.*

p8 line 15: "GEBCO" Which version of GEBCO and at which resolution.

*See answer to the first comment in the previous section.*

p9 line 2: "reports" considered illustrates instead

*Ok*

P9 line 14: "The picture of Figure 5" consider "The wave height distribution illustrated in Figure 5" instead

*Ok*

p9 line 16: "ranges" you mean reaches?

*Ok*

P10 line 2: "stretch" consider using area instead

*Ok*

Figure 5: "in the legend together with the inundation distance (I) and runup (R)", I would add the word "observed" to avoid any misunderstanding, "in the legend together with the observed inundation distance (I) and runup (R)"

*Correct remark, the word "observed" has been added in the figure caption.*

p13 line 7: It seems that the eastern most extreme is reached after 40s (i.e. T180), although it depends on where you put the limit for eastern extreme.

*Ok*

p13 line 8: "Contemporarily" consider at the same time instead.

*ok*

p13 line16: consider illustrated instead of "reported"

*Ok*

p13 line 17: "(#1)" consider (#1 figure 5) instead. Is this point #1? It Is not clear.

*Indeed, it's Grid 2. Fixed.*

p14 lines 3-9: Some refinement in English language will improve the text.

*Done.*

p14 lines 11-18: Some refinement in English language will improve the text.

*Done.*

p14 line 21: "agents" consider factors instead

*"agents" refers to atmospheric manifestations (rain, wind, storms and so on), that are usually denoted with this word.*

p14 line 22: "Basing" consider Based instead

*Ok*

p15 line 5: "The correction done is shown in green-blue when negative (meaning "digging" with respect to the present ground level) and in yellow-red when positive, meaning increased ground elevation." Some refinement in English language will improve the text.

*Done*

p16 lin2: "reported" consider shows or illustrates instead.

*Done*

p16 line7 : "The most relevant changes regard the area between the north-east corner of Pantano Piccolo and the Torre Bianca site, where a 1 to 5 m surface layer of ground has been removed." Consider "The

most relevant change regarding the area between the north-east corner of Pantano Piccolo and the Torre Bianca site is the removal of 1 to 5 m of surface ground layer.

*Done*

p16 line 11: "agents" consider factors instead

*Same as above*

p17 line2: "filed" consider frame instead

*Ok*

p17 line 10: "chief" consider main instead

*Ok*

p19 line 4: "outside" consider besides instead

*Ok*

**AUTHORS ANSWERS TO REFEREE #2's COMMENTS**

Zaniboni et al. investigate the Scilla tsunami, which is one of the deadliest tsunamis ever occurred in Italy. They simulated the tsunami generated by the subaerial landslide that followed one of the largest aftershocks that occurred during the 1783 earthquakes storm in Calabria, Italy, and compared the results against the historical evidences. Since there were still some disturbing gaps between the computed results and the historical evidences, they improved the grid resolution and restored the topography of the Pantano Piccolo area in northeastern Sicily at the time of the tsunami. This way they were able to match very nicely the computed results with the actual evidences.

This is an elegant investigation, relevant to nowadays tsunami hazard research and evaluation. The study is well constructed, conducted very carefully with much attention to the fine details. The understandings extrapolate the past experience onto the future, emphasizing the hazard posed by tsunamis induced by subaerial landslides, the need to stay away from the coast after strong shaking, and the vulnerability of coastal channels to tsunami penetration inland.

The manuscript is certainly suitable for NHESS, but I would recommend some corrections and improvements before publication as follows:

General Comments

Abstract:

I suggest the authors to state and stress the importance of morphological reconstruction they have implemented in the grid around the area of Pantano Piccolo in order to achieve a better agreement between the high resolution scenario computation and the past evidences.

*We agree with this suggestion, a sentence has been added to the abstract.*

Conclusions:

Rather than a focused, short and concise, this section is a mixture of results, discussion, summary and some conclusions; it is of the longest sections in the manuscript and hard to follow. For example, the first paragraph is a summary of Zaniboni et al. (2016), the second paragraph is a summary of the present study, the bullets section is "main results and findings" (P. 18 line 18), the middle bullet in page 19 is mainly discussion, etc; and the actual conclusions are spread along this section and hard to follow. I suggest reorganizing the Conclusions section, it can be divided into discussion and conclusions, or any other useful way; it should be shorter with less repetitions of what have already been said before. It would also be useful to refer back to the relevant figures, and this will help the reader to follow the mentioned issues. At the moment there is only one as such reference (P. 18, line 11). I am not an English speaking person, but had the feeling that some language editing is needed.

*The Conclusions section has been reorganized following this advice. A Discussion section has been added, resuming and discussing the main results, while the Conclusions section has been shortened.*

Technical comments:

P. 1, Line 12: Please consult the editor whether to use a reference in the abstract

*The reference has been removed.*

P. 1, line 14: ": : : three tsunami simulations." while P. 2, lines 11-12 mention two tsunamis: : :;

*Ok, fixed.*

P. 1, line 17: Would be more accurate to say 'regional' rather than 'global';

*Ok*

P. 6, line 27: Do you mean 'seismogenic' rather than 'tsunamigenic'?

*Changed "tsunamigenic failure" to "tsunamigenic source" (Page 6, Line 27 of the new version).*

P. 9, line 2-3: 'cleared out' means to empty, remove, leave, etc: : : I believe you mean 'will be explained'? If so, please rephrase.

*Done (Page 9, Line 6 of the new version).*

P. 14, Lines 8-9: Please indicate which of the simulations is not in line with the historical accounts.

*The sentence is referred to both simulations, on Grid 2 and Grid 3, that are named just before. The expression "the simulations" has been substituted with "both simulations" (Page 14, Line 1 of the new version).*

P. 15 and 17, Captions of Figures 9 and 11: While referring to the upper and lower panel, use colon ":" rather than right side bracket ")".

*Done.*

Figure 1: The study area is quiet familiar to the Mediterranean and the European communities. However, I would suggest the introducing of an inset that gives a wider geographic orientation for those around the world who are not familiar with this region.

*A larger map of Italy, with the indication of the studied area, has been added.*

Figures 2 and 5: Please add the location of Pantano Piccolo and San Saba that are mentioned later on in the text.

*Figure 2 illustrates the location of the observed effects, while the ones in San Saba are obtained from the numerical simulations. The toponym has been added to Figure 5. The position of Pantano Piccolo is reported in Figure 6. Figures 2 and 5 cover a wider region and adding Pantano Piccolo would create confusion.*

Table 1: There were several tsunamis in Calabria and Sicily during the 1783 earthquakes crisis. Please mention in the caption the exact date of the Scilla tsunami;

*Done*

Please verify whether the historical sources were careful enough to differentiate the effects induced by the Scilla tsunami from the effects of the other tsunamis (e.g. the February 5, 7, March 1, 28);

*The main tsunami evidences (excluding the February 6th, 1783 one) are associated to the stronger earthquake of the day before, producing significant waves on the coasts of Sicily and Calabria. Their effects are very well reconstructed and reported in the paper by Graziani et al. (2006), clearly distinguishing the main features of the different events, basing on detailed historical reports.*

In order to get a comprehensive perspective of the impact of the Scilla tsunami, I would suggest to complete the given list. For example, please mention what had happened in Pantano Piccolo, San Saba, and elsewhere if known. In my opinion, it worth mentioning also what had happened in Scilla even though this was already investigated in the previous, 2016 paper;

*The effects in the Scilla beaches have been recalled in the text (Section 2, describing the 6th February 1783 tsunami effects). Thus we preferred to avoid repetition in Table 1.*

Punta del Pezzo: Please verify whether the sea affected one and a half mile stretch of the beach. One may think that the sea inundated one and a half mile into the land?

*The Calabrian coast is very steep, with narrow beaches and high topographical gradients. So, in that case a similar inland penetration is not possible (see also topography in Figure 6)*

**AUTHORS ANSWERS TO REFEREE #3's COMMENTS**

This manuscript describes new analysis of previous studied 1783 Scilla landslide. The analysis in this paper is focused on the inundation of the generated tsunami in the areas close to the landslide. It also includes a reconstruction of the topography to fulfil the historical observations.

The paper is very interesting, and I liked the open and transparent way the results are presented. Good and clear figures. I have suggested minor revision with some suggestions to improvements below.

Main comments:

1) Discussion of results of simulations on Grid2. Line 16 on page10: Looking at the maximum surface elevations of Fig. 5, the main argument for not achieving observed runup at location 5 and 8 is the resolution of the grid. Why not perform grid refinement tests? I guess that the resolution is high enough for propagation in the deepest part of the strait. 0.3 m before runup cannot give 2 and 3.2 m runup.

*Grid refinements in the harbours of Reggio Calabria (#5) and Messina (#8) could account for higher waves, associated to the tsunami itself and to resonance phenomena. This very interesting task has not been explored yet, and it is indicated as one of the topics deserving attention for a future work.*

2) What is the original resolution of the data SRTM and GEBCO used in this paper

*A more detailed description of the original datasets and of their use to obtain the computational grids have been added (page 8 lines 15-19).*

3) Grid refinement tests – should be shown or at least referred to for all grids (not only the 50 m grid, but in sea and on land for 10 m grid)

*The grid refinement has been done and discussed only for the area of Capo Peloro, showing the improvements brought by such procedure. It can then be applied to other areas of interest, where detailed historical reports are available, such as Messina and Reggio Calabria harbors as well as the coastal area of Punta del Pezzo, about 7 km west of Scilla along the Calabrian coast. As already stated above, this is left to further studies.*

Minor comments

1) Table 1 – must have a ref. to Fig 2

*Done*

2) Line 1 page 7: what is "cellular automata"

*It's a simulation technique, described and applied in the related paper, that is not worth of description here.*

3) L5 p7: check super scripts: m3->m^3, m2->m^2 etc. Check entire paper

*Ok*

4) L8 p7 vs L18 p7. Inconsistent use of "million" and "M". Check entire paper

*Ok*

5) Fig 3: a vertical line at shoreline will help reading the figure. Include also the location of "blocky deposit", not only simulated deposits

*A horizontal light-blue line has been added, marking the sea level. A profile of the blocky deposit is not available*

6) Fig 3: Use of only end-parentheses for dividing the text for different panels. I think it is better to use colon (Check rest of paper)

*Ok*

7) L8 p8: Higher grid resolution give more accurate results must be more discussed. Resolution of grids and data, stability etc.

*As already mentioned above, these issues have been discussed for the Capo Peloro area. Such details are interesting but too specific for the aims of this work.*

8) L14 p9: the sentence starting with "The picture of : : :" must be revised – I could not understand what was meant here

*The sentence has been slightly modified (Page 10, Lines 4-5 of the new version)*

9) Chap 5.1 – I think some mariograms for 3-4 locations also could be fine for better understand the wave pattern

*We decided to omit the virtual marigrams obtained by the simulations for two main reasons: i) there are not instrumental records to compare with; ii) they did not add particularly interesting information to the already shown plots (propagation and maximum water elevation).*

10) Fig 6: include also depth toward location 5 and 8.

*This remark is not particularly clear: the picture shows the area covered by Grids 3 and 4, not including the harbors of Messina and Reggio Calabria, located southward.*

11) L19 p 11: what is meant by "this zone"?

*No "this zone" exists at the indicated line and page. There is "coastal zone" (at Page 11, Line 24 of the new version), that is clearly referred to the Sicilian region, already cited twice in the same sentence and not repeated.*

12) L21 p11. Revise sentence "If in Calabria: : :". Show the lowland of Capo Peloro in a map?

*The Capo Peloro lowland and Calabria coasts are shown clearly in Figure 6.*

13) L18 p13: What is meant by (#1), similar L32 p18 (#2). Is it "Grid 2"?

*To avoid continuous repetition of the word "grid", sometimes they have been referred to only using their number ("#1" is for Grid 1, "#2" for Grid 2 and so on).*

14) Fig. 11 upper panel. For better comparison to Fig 8, use same scales!

*Correct remark, the water height scale and the color palette of Figure 8 in the new version match those of Figure 11.*

15) L12-14 p18: Check sentence

*Done.*

16) L30 p18: "inundation does not fit observations"??? See L12 p18 and bulletpoint at L28 p19 and elsewhere where you have concluded that the simulations is a good reconstructions.

*The three reported sentences refer to three different contexts: the first concerns the 50-m grid simulation, not fitting the observed run-up and then justifying grid refinement; the second refers to the simulation of the 10-m grid in Scilla beaches, presented in the previous paper (Zaniboni et al., 2016); the third one discusses the good agreement reached with the morphology reconstruction adopted for Grid 4.*

17) L6 p19: "basing" use based instead?

*Ok, fixed.*

18) L12 p19. What is meant by "better resolution" – finer grid or higher resolution of the data

*It means finer grid, the text has been changed accordingly.*

**Assessment of the 1783 Scilla landslide-tsunami effects on Calabria and Sicily coasts through numerical modeling**

Filippo Zaniboni[1,2], Gianluca Pagnoni[1], Glauco Gallotti[1], Maria Ausilia Paparo[1], Alberto Armigliato[1], Stefano Tinti[1]

[1]Dipartimento di Fisica e Astronomia, Università di Bologna, Bologna, Viale Berti-Pichat 6/2, 40127 Bologna, Italy
[2]Istituto Nazionale di Geofisica e Vulcanologia – Sezione di Bologna, Bologna, via Donato Creti 12, 40128 Bologna, Italy

*Correspondence to*: Filippo Zaniboni (filippo.zaniboni@unibo.it)

**Abstract.** The 1783 Scilla tsunami, induced by a coastal landslide occurring during an intense seismic sequence in Calabria (South Italy), was one of the most lethal ever observed in Italy. It caused more than 1500 fatalities, most of which on the beach close to the town where people gathered to escape earthquake shaking. In this paper, complementing a previous work focusing on the very local tsunami effects in the town of Scilla, we study the tsunami impact on the Calabria and Sicily coasts out of Scilla. To this purpose we take into account the same landslide geometry considered in the previous study and perform three tsunami simulations, one embracing a larger region with a 50-m computational grid, and two covering the specific area of Capo Peloro, in Sicily, facing Scilla on the western side of the Messina Straits, with even higher resolution (10 m mesh). In this area, reconstructing the local morphology at the time of the tsunami occurrence allows to account for the inundation of the small lake of Pantano Piccolo, event reported in historical records but not reproduced in the previous numerical simulations. In general, the 
[revised manuscript text omitted]

[Figure]

**Figure 1: The logo of Copernicus Publications.**